# Improvement in the decadal prediction skill of the North Atlantic extra-tropical winter circulation through increased model resolution

Mareike Schuster[1], Jens Grieger[1], Andy Richling[1], Thomas Schartner[2], Sebastian Illing[1], Christopher Kadow[1], Wolfgang A. Müller[4], Holger Pohlmann[3,4], Stephan Pfahl[1], and Uwe Ulbrich[1]

[1]Freie Universität Berlin, Institut für Meteorologie, Carl-Heinrich-Becker Weg 6-10, 12165 Berlin
[2]Deutscher Wetterdienst, Güterfelder Damm 87-91, 14532 Stahnsdorf
[3]Deutscher Wetterdienst, Bernhard-Nocht-Straße 76, 20359 Hamburg
[4]Max-Planck-Institut für Meteorologie, Bundesstraße 53, 20146 Hamburg

**Correspondence:** Mareike Schuster (mareike.schuster@met.fu-berlin.de)

**Abstract.** In this study the latest version of the MiKlip decadal hindcast system is analyzed and the effect of an increased horizontal and vertical resolution on the prediction skill of the extra-tropical winter circulation is assessed. Four different metrics - the stormtrack, blocking, cyclone and windstorm frequencies - are analyzed in the North Atlantic and European region. The model bias and the deterministic decadal hindcast skill are evaluated in ensembles of 5 members in a lower resolution version (LR, atm: T63L47, ocean: 1.5° L40) and a higher resolution version (HR, atm: T127L95, ocean: 0.4° L40) of the MPI-ESM system. The skill is assessed for the lead winters 2-5 in terms of the anomaly correlation of the quantities' winter averages, using initializations between 1978 and 2012. The deterministic predictions are considered skillful, if the anomaly correlation is positive and statistically significant. While the LR version shows common shortcomings of lower resolution climate models, e.g. a too zonal and southward displaced stormtrack and a negative bias of blocking frequencies over the eastern North Atlantic and Europe, the HR version counteracts these biases. Especially cyclones, i.e. their frequencies and characteristics like strength and lifetime, are better represented in HR. As a result, a chain of significantly improved decadal prediction skill between all four metrics is found with the increase of the spatial resolution. While the skill of the stormtrack is significantly improved primarily over the main source region of synoptic activity - the North Atlantic Current - the other extra-tropical quantities experience a significant improvement primarily downstream thereof, i.e. in regions where the synoptic systems typically intensify. Thus, the skill of the cyclone frequencies is significantly improved over the central North Atlantic and Northern Europe, the skill of the blocking frequencies is significantly improved over the Mediterranean, Scandinavia and Eastern Europe and the skill of the windstorms is significantly improved over Newfoundland and Central Europe. Not only is the skill improved with the increase in resolution, but the HR system itself exhibits significant skill over large areas of the North Atlantic and European sector for all four circulation metrics. These results are particularly promising regarding the high socio-economic impact of European winter windstorms and blocking situations.

# 1 Introduction

The extra-tropical circulation plays an important role for the redistribution of energy in the atmosphere. The prevailing wester-
lies and the embedded cyclones and anticyclones determine the weather and climate of the mid-latitudes, assisting in balancing
temperature and humidity contrasts between tropical and polar regions. Natural climate variability as well as externally forced
climate change determine fluctuations in the circulation and thus i.a. in the frequency of extremes, such as strong cyclones,
intense windstorms or phases of blocked flow. The consequences of such features include extremes in temperature, precipita-
tion/drought and wind speed, often accompanied by immense damage (e.g. Leckebusch and Ulbrich, 2004; Ulbrich et al., 2009;
Sillmann and Croci-Maspoli, 2009; Pfahl and Wernli, 2012; DeutscheRück, 2018). Therefore, the societal demand for reliable
near term climate predictions of such features - also to support political, economical and administrative decision making - is
perpetually growing.

Decadal Climate prediction is an active research field in climate sciences. Different research groups around the globe aim
at the development of skillful prediction systems (Boer et al., 2016). Retrospective forecasts, termed hindcasts, are used to
assess the ability of the model systems to predict climate variability on inter-annual to decadal time scales. Initialized from
observation-based data and run for a period of 10-30 years, decadal climate predictions combine forecast elements from weather
and seasonal forecast divisions (initial conditions) as well as from long-term climate projections (boundary conditions). To date,
different designs of decadal prediction systems are prevalent. They either consist of a multi-member single-model suite, such
as the UK MetOffice's Decadal Prediction System 'DePreSys' (Smith et al., 2007) and the German 'Mittelfristige Klimaprog-
nosen (MiKlip) system' (Marotzke et al., 2016), or they are based upon a multi-model suite, as used e.g. within the 5[th] Coupled
Model Intercomparison Project (CMIP5; Taylor et al. 2012). Several multi-model studies, using the CMIP5 decadal prediction
suite, come to the conclusion that there exists significant prediction skill on decadal time scales (e.g. Kim et al., 2012; Doblas-
Reyes et al., 2013; Meehl et al., 2014). Results from these studies have also been included in the 5[th] assessment report (AR)
of the Intergovernmental Panel on Climate Change (IPCC; Kirtman et al. 2013). Currently, in preparation for the 6[th] AR of the
IPCC and CMIP6, improved decadal prediction systems are developed (Eyring et al., 2016; Boer et al., 2016; Kushnir et al.,
2019).

With respect to atmospheric quantities, various versions of the 'MiKlip system', based on the Max-Planck Institute Earth
System model (MPI-ESM), show decadal prediction skill mainly in terms of global and regional temperature indices (Müller
et al., 2012; Pohlmann et al., 2013; Kröger et al., 2017). The skill for precipitation, due to its complex and partly small-scale
nature, is however regionally confined and prevalently limited to lead year 1 (Kadow et al., 2016). Results of practical relevance
within MiKlip are on the one hand skill for the prediction of 10 m wind speed and wind energy over Central Europe, which
exists irrespective of the ocean initialization technique (Moemken et al., 2016) and on the other hand skill for the prediction of
wind speeds of different quantiles upward from 75% (Haas et al., 2016).

Kruschke et al. (2014) and Kruschke et al. (2016) analyzed the forecast skill of Northern Hemisphere cyclone and windstorm
frequencies in the MiKlip system, respectively. They found probabilistic decadal forecast skill for cyclone frequencies in some

areas over the Northern Hemisphere ocean basins, mainly the North Sea and the central Pacific, whereupon the sub-sample of strong cyclones exhibits generally higher skill than the complete sample of all cyclones (Kruschke et al., 2014). Using a parametric bias adjustment approach, Kruschke et al. (2016) found windstorm frequencies for winters 2–5 and winters 2–9 to be skillful over large parts of the Northern Hemisphere when compared against climatological forecasts. Another study

using the MPI-ESM decadal prediction system demonstrated that the decadal prediction skill for surface temperature and cyclone frequencies can be significantly improved by replacing each ensemble member's ocean state with the ensemble mean ocean state at regular intervals during the forecast period (Kadow et al., 2017). More studies using the MPI-ESM hindcasts are collected in a special issue of the Meteorologische Zeitschrift about the validation of the MiKlip system in its first phase (Kaspar et al., 2016).

It is well known that a coarse spatial resolution of global coupled climate models hinders the proper representation of sub-synoptic scale systems, and thus the climate mean state and variability. For example, with respect to the North Atlantic and European domain many lower resolution climate models exhibit a cold sea surface temperature (SST) bias south of Greenland, due to a displacement of the North Atlantic Current or a too weak overturning circulation (Park et al., 2016; Scaife et al., 2011; Wang et al., 2014). This common bias in the North Atlantic Current is associated with a too zonal stormtrack, stronger

geopotential height gradients in the mid-latitudes, increased westerlies and reduced blocking frequencies over Europe (e.g. Scaife et al., 2011). It has been found in many studies that the atmospheric dynamics benefit not only from a coupling of the atmosphere and ocean but also from an increased model resolution (e.g. Shaffrey et al., 2009; Jung et al., 2012; Dawson et al., 2013; Hewitt et al., 2016). Scaife et al. (2011), for example, demonstrated that the increase in resolution in both the atmospheric and oceanic model components results in a chain of improvements, as they found a reduced SST bias in the higher resolution

model which in turn lead to a better representation of westerly winds and blocking frequencies. Similar effects can also be found in atmosphere only models, e.g. for the blocking frequency bias in Davini et al. (2017).

   With an atmospheric resolution of T63L47 (about 1.875° horizontal grid spacing) and an oceanic resolution of 1.5°L40 the MPI-ESM-LR decadal prediction system applied in the first phase of MiKlip has a rather moderate spatial resolution. Meanwhile, studies using higher resolution forecast systems are available, for instance Monerie et al. (2017) using 0.5° grid spacing

in the atmosphere and 0.25° in the ocean and Robson et al. (2018) using ∼0.9° in the atmosphere and 0.25° in the ocean. They focus on oceanic parameters and find skill e.g. for SSTs, sea ice extent and ocean heat content, respectively. However, systematic analyses of the actual effect of the increase in resolution on the hindcast performance on the decadal scale are rare. Pohlmann et al. (2013) found for the hindcasts of mixed resolution (MPI-ESM-MR) that an increase in vertical (atmosphere: T63L95) and horizontal resolution (ocean: 0.4°L40) compared to MPI-ESM-LR improves the tropical Pacific surface tempera-

ture predictions in the lead years 2-5 and leads to a good representation of the quasi-biennial oscillation (QBO), which remains in alignment with observations well beyond the first 12 months after initialization. Apart from that, the mixed resolution shows only modest benefit for the hindcast skill (Marotzke et al., 2016).

   In this study, for the first time an analysis of the direct impact of the model resolution on the skill of decadal climate predic-

tions of dynamical variables is performed under otherwise unchanged model settings (parametrization and initialization). We

evaluate the MiKlip hindcasts performed with the latest version of the Max-Planck Institute Earth System model with higher resolution (MPI-ESM-HR, Müller et al. 2018), which will contribute to CMIP6, and compare its decadal forecast skill to that of a previous lower-resolution version (MPI-ESM-LR). While many studies analyzing the skill of decadal forecast systems tend to focus on basic atmospheric variables such as the surface temperature and precipitation (e.g. Smith et al., 2007; Keenly-side et al., 2008; Goddard et al., 2013; Kadow et al., 2016; Monerie et al., 2018; Xin et al., 2018), we emphasize the role of dynamical processes and therefore analyze a set of quantities representing the extra-tropical winter dynamics: the stormtrack, blocking, cyclones and windstorms.

We introduce the MPI-ESM prediction system as well as the skill measure used to assess the hindcast quality in Sect. 2.1. In Sect. 2.2 we describe the different circulation quantities in detail and present their climatology in the ERA-Interim reanalysis with a focus on the North Atlantic and European region. The model climatologies and biases are discussed in Sect. 3.1 and the prediction skill of the winter circulation is evaluated in Sect. 3.2. In Sect. 4 we discuss and relate our findings to other studies, before we conclude our results in Sect. 5.

## 2 Data and methodology

The extra-tropical circulation in the Northern Hemisphere is most active during the winter season, with a stronger jet stream in the upper-troposphere and numerous strong cyclones developing in the mid-latitude baroclinic areas, favored by strong horizontal temperature contrasts resulting from relatively warm ocean currents near the surface and cold polar air masses. Storms that strike the European continent at this time of the year are often powerful and damaging. We will therefore focus on the winter circulation and evaluate averages of the stormtrack and blocking, cyclone and windstorm frequencies from October through March. The stormtrack describes the variability of baroclinic waves on synoptic time-scales in the extra-tropics. These baroclinic waves are a combination of two contributing components, i.e. anti-cyclonic and cyclonic anomalies, which we will analyze in terms of blocking frequencies on the one hand and extra-tropical cyclone and windstorm frequencies on the other hand.

To assess the model bias and to compute the prediction skill of the different diagnostics in the decadal hindcasts, a reference (i.e. observational) data set is needed. However, there exists no gridded observational dataset for the metrics of interest. Instead we make use of a reanalysis product and derive the circulation quantities for the winters 1979/80 to 2016/17 from the ERA-Interim reanalysis (Dee et al., 2011), created by the European Centre for Medium-Range Weather Forecasts (ECMWF), with a horizontal resolution of T255 ($\sim$0.75°) on 60 levels and a top of the atmosphere at 0.1 hPa.

### 2.1 Forecast system and skill measures

The two decadal forecast systems that we compare are both based on the Earth System Model of the Max-Planck-Institute for Meteorology (MPI-ESM) version 1.2, which is a coupled atmosphere ocean model and consists of the atmospheric component ECHAM6.3 and the oceanic component MPI-OM1.6.2. The lower resolution of MiKlip's pre-operational decadal prediction

system (MPI-ESM-LR, termed LR hereafter) has an atmospheric horizontal resolution of T63 (1.875°) and 47 levels, with the top of the atmosphere at 0.01 hPa (Mauritsen et al., 2019). The ocean component is run with 1.5° L40. A general skill assessment of decadal predictions performed with the LR system can be found in Polkova et al. (2019). The higher resolution version (MPI-ESM-HR, termed HR hereafter) uses T127 (0.9375°) and 95 vertical levels for the atmosphere, and 0.4° L40 for the ocean (Müller et al., 2018). The HR version therefore has a finer grid in both the atmosphere and the ocean components. For this analysis, both systems use the CMIP5 external forcing with respect to greenhouse gases and aerosols (for details see Giorgetta et al. 2013). Both systems are full-field initialized in the atmosphere, using ERA-40 (Uppala et al., 2005) and ERA-Interim (Dee et al., 2011); and anomaly-initialized in the ocean, using ORA-S4 (Balmaseda et al., 2013) and sea-ice concentration from the National Snow and Ice Data Center (NSIDC). The initialization procedure is identical to the one used for MiKlip's Baseline1 system and is described in more detail in Pohlmann et al. (2013). The LR system consists in total of 10 ensemble members, initialized annually between 1960 and 2016, with each initialization covering one decade. The integration period for each of the initializations spans 10 years. However, since the HR system - with an otherwise identical hindcast setup - consists of only 5 members, and to guarantee a fair comparison between the two forecast systems we only evaluate the first 5 members of LR as well.

To derive the deterministic skill of the two forecast systems, we focus on the temporal variability and analyze the anomaly correlation for the winters 2-5 (Oct-Mar), following the Decadal Climate Prediction Project (DCPP, Boer et al. 2016) protocol. That means that we calculate lead time dependent anomalies of the circulation measures. This is a simple and robust approach to account for a possible lead time dependent mean bias, i.e. drift. Thus, for each of the initialization experiments (1978, 1979, ...) the ensemble average (5 members) of the temporal mean of the 4 contained lead winters is calculated per grid point. This forms a new ensemble mean time series of the lead winters 2-5. This time series serves to calculate the climatology (temporal mean) as well the respective anomaly time series. The time series of those anomalies of the hindcasts is then correlated (Pearson) to the time series of anomalies of the reanalysis. In decadal prediction studies, this procedure is usually repeated for each lead time, e.g. lead year 1, lead year 2-5, lead year 6-9 - it is therefore referred to as lead time dependent anomaly correlation. In our study we only show results for one lead time: lead winters 2-5. The initialization of the hindcasts takes place in October, this means the first full winter that we analyze is the second winter, i.e. the months 12-17 (Oct-Mar) after initialization. This evaluation procedure is part of the decadal climate prediction evaluation software that was designed within the MiKlip project (Illing et al., 2014) and is applied for this study. This OpenSource evaluation software follows the evaluation framework of Goddard et al. (2013) which led to the DCPP requirements.

To match the period covered by the ERA-Interim reanalysis, we do not use the full set of initializations but instead use the decadal hindcast experiments that are initialized between 1978 (winter 2: 1979/80, winter 5: 1982/83) and 2012 (winter 2: 2013/14, winter 5: 2016/17) in LR and HR. In total we therefore analyze 700 October-to-March winter seasons (5 members x 35 initializations x 4 lead winters) per forecast system. The skill of each of the forecast systems (LR, HR) is first evaluated against the reanalysis data, i.e. the anomaly correlation between the respective hindcast and ERA-Interim is determined. Then, the two systems are compared against each other, i.e. the difference of the aforementioned correlations between the two forecast systems is computed. To determine the significance of the correlation (95% significance level), the time series of reanalysis-

**Figure 1.** Climatology of the winter average (Oct-Mar) of different circulation quantities in the ERA-Interim reanalysis for the period 1979/80-2016/17. The stormtrack, i.e. the standard deviation of the 500 hPa geopotential height anomaly is shown in m (45-60 by 5). The fraction of blocked days is shown in % (4-8 by 2). The cyclone frequency (120-180 by 20) and windstorm frequency (25-30 by 2.5) are shown in number of tracks within a radius of 1000 km. Grey masked areas denote grid points with an orography larger than 1500 m, which have been omitted for cyclone identification.

hindcast pairs is resampled with replacement 1000 times (block bootstrap taking auto-correlation into account), following Goddard et al. (2013).

## 2.2 Circulation metrics

### Stormtrack

5 The extra-tropical stormtrack is derived from the bandpass filtered variability of the geopotential height at 500 hPa in the window of 2.5 to 6 days, an Eulerian approach following Blackmon et al. (1976). Its long term winter average (October through March) is displayed in Fig. 1 for the North Atlantic and European region and the period 1979/80-2016/17 based on the ERA-Interim reanalysis. The North Atlantic stormtrack is visible in green shades, with its maximum of 60 m located over the western North Atlantic and Newfoundland and a typical north-eastward tilt.

### Blocking

For atmospheric blocking a slightly modified version of the 2-dimensional blocking index of Scherrer et al. (2006), based on gradients in the daily 500 hPa geopotential height field, is used to identify instantaneously blocked grid points. In contrast to Scherrer et al. (2006), where a blocking area is defined in between the blocking high and the associated low, here the position of

15 detected blocked grid points is shifted north by $7.5°$ to correspond better with the anticyclonic part of a blocking situation. To account for large-scale and persistent blocking anticyclones between $35°N$ and $80°N$, an adapted tracking algorithm for blocking regimes, similar to the approach by Barnes et al. (2012), is applied. With this tracking method, we only select contiguously blocked regions with a minimum zonal and meridional extension of $\sim 15°$ and an area of at least $1.5 \times 10^6$ km$^2$ lasting for a minimum of 4 days. A possible shifting, merging and splitting of blocking areas in time is considered by adopting a blocking

overlap area criterion of 750.000 km$^2$ between two consecutive days and a maximum distance between blocking centers of 1000 km. The climatology of the mean winter blocking frequency is displayed in blue isolines in Fig. 1. Its maximum of 8% blocked days stretches from the Azores to Scotland. A second region of increased blocking frequencies is found between Greenland and Iceland.

### Cyclones

To identify and track extra-tropical cyclones we apply an objective Lagrangian feature tracking algorithm, developed by Murray and Simmonds (1991), to 6-hourly values of the mean sea level pressure. Maxima of the Laplacian of the mean sea level pressure are identified and, if a minimum in the pressure field itself can(not) be detected in the vicinity, a closed (open) cyclone

is identified. The system is then tracked in time, at 6-hourly time steps. Only cyclones that live for more than 24 hours and reach a Laplacian of pressure larger than 0.7hPa/(degree latitude)$^2$ and have closed isobars at least once during their lifetime are selected for evaluation. The measure we ultimately use for our evaluation is the cyclone frequency, i.e. the number of cyclone tracks that pass within a radius of 1000 km of the respective grid point on a 2.5° x 2.5° grid. As the extrapolation of pressure to sea level can be erroneous over high terrain, cyclones are not identified at grid points where the orography is higher than

1500 m. The winter average of the cyclone frequency is displayed in Fig. 1 in red dashed contours. Its maximum is located at the southern tip of Greenland with 180 cyclones and a band of enhanced cyclone frequencies is located downstream of the stormtrack maximum with a similar southwest-northeast tilt.

### Windstorms

Yet another objective Lagrangian tracking scheme is used to derive the frequency of extra-tropical windstorms (Leckebusch et al., 2008; Kruschke, 2014). This method is based on the exceedance of the local 98th percentile of the near-surface wind speed to define contiguous fields of strong wind. Percentiles are calculated for each model simulation (LR, HR) and the reanalysis individually, using 6-hourly data of the whole year between 1981 and 2010. For the hindcasts the percentiles of the uninitialized counterparts are used as done by Kruschke et al. (2016). Windstorms are identified if the area of wind exceedance

above the percentile is larger than 150.000 km$^2$ and if the feature is trackable for at least 18 hours. Tracking is done by means of a nearest neighbour approach. The individual windstorm tracks are further used to calculate windstorm frequencies, which are computed identically to those of the cyclone frequencies. The yellow dotted contours in Fig. 1 represent the average winter windstorm frequency. Its maximum of 30 windstorms is located also downstream of the stormtrack maximum, but slightly shifted southward compared to the cyclone frequencies. This illustrates that the corresponding windstorm field is usually lo-

cated to the south of the cyclone center, where the pressure gradients and thus geostrophic wind velocities are typically largest.

The software routines that were used to compute all the extra-tropical circulation quantities, as well as the evaluation procedure were implemented as separate plug-ins into the MiKlip Central Evaluation System (https://www-miklip.dkrz.de) - based on the Free Evaluation System Framework (Freva, Kadow et al. in review) - by their developers and authors of this paper.

The single plug-ins and their documentation can be found under https://www-miklip.dkrz.de/plugins - plus the respective suf-

fix /stormtrack/detail/ (Stormtrack); /blocking_2d/detail/ (Blocking); /zykpak/detail/ (Cyclones); /wtrack/detail/ (Windstorms) and /murcss/detail/ (skill analysis).

## 3 Results

### 3.1 Model bias

Before we analyze the decadal prediction skill, we will first evaluate the ensemble mean climatology and bias, in order to assess the model's capability to represent the four atmospheric circulation features. For this, we only take into account those seasons that will be used for the skill analysis, i.e. the winters 2-5 of each of the 35 initializations (1978-2012) and 2 x 5 members (LR, HR). To compute the model bias, we consider the entire reanalysis data set, i.e. winters from 1979/80 to 2016/17.

In Fig. 2 the model bias for the stormtrack and blocking frequency compared to ERA-Interim is displayed in colored shades and the respective model climatology is shown in grey contours, for both the LR and HR ensemble mean. The grey contour levels are the same as for the ERA-Interim climatology in Fig. 1. The LR system shows the typical North Atlantic stormtrack along 45°N, with a maximum over the western part of the basin, however rather zonally aligned and shifted southward (Fig. 2a). Since the observed stormtrack is tilted from south-west to north-east (see Fig. 1), this results in a negative bias (-10m) at
higher latitudes and a positive bias (+8m) at lower latitudes in the LR prediction system. This bias can partly be corrected with the increase in the model resolution, as the HR system increases the stormtrack activity where there is a negative bias in LR and vice versa (Fig. 2c), however this effect is strongest at the northern side of the stormtrack, as also seen in Müller et al. (2018). In HR the North Atlantic stormtrack is more tilted, and therefore closer to observations (Fig. 2b). Not only does it extend further north in the higher resolution system, but it also extends further downstream towards Central and Eastern Europe, and
therefore reduces the negative bias over the North Sea and Scandinavia that is present in LR. The bias in HR is reduced at both the northern and southern flanks of the Atlantic stormtrack, however the southward shift over the central North Atlantic is still present (-7m and +7m).

The blocking frequency shows a negative bias of fraction of blocked days per winter (-3%) in the LR system just north of its
climatological maximum, i.e. over a band stretching from the central North Atlantic and Great Britain towards the Baltic Sea, and a positive bias (+1.5%) over the Mediterranean (Fig. 2d). Fig. 2f illustrates that again the HR prediction system counters these shortcomings of the LR system and reduces the bias in the right places, but the effect is rather marginal for this quantity. Though weaker, the bias of the blocking frequency in HR is still considerable (-2.5% and +1% respectively). These findings are in line with the analysis of blocking in Müller et al. (2018).


The climatology of the cyclone frequency with its maximum at the southern tip of Greenland, seen in Fig. 1, is also visible in LR (Fig. 3a). In contrast to the stormtrack and blocking frequency, the cyclone frequency in LR does not exhibit a clear southward shift compared to the reanalysis. Instead, in the low resolution system there are overall far too many cyclones present

## Stormtrack

## Blocking Frequency

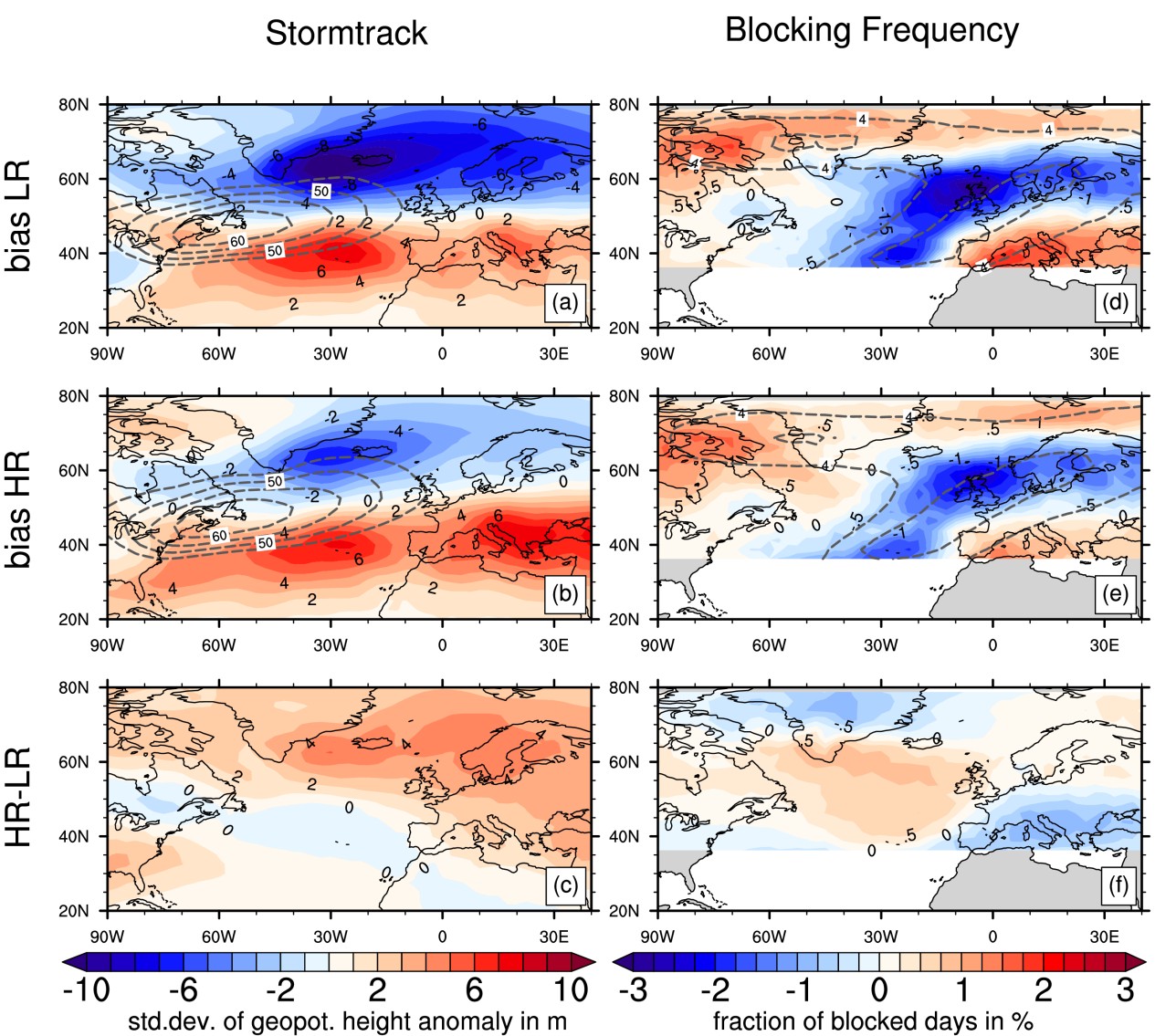

**Figure 2.** Ensemble mean model bias relative to ERA-Interim (shading) and model climatology (dashed contours) of the respective circulation quantity in LR (top row), HR (middle row) and the difference between HR and LR (bottom row). The circulation quantities displayed are the stormtrack (left) and the blocking frequency (right). Initializations from the period 1978-2012 are used for 5 members of each, LR and HR, and the ensemble mean is computed from lead-time averages over the hindcast winters 2-5 (Oct-Mar). In ERA-Interim the winters between 1979/80 and 2016/17 are used. The grey contours, i.e. ensemble mean climatology, have the same levels as in Fig. 1 - 45-60 by 5 m for the stormtrack and 4-8 by 2 % for the blocking frequency.

between 30°-70°N, but especially over the central North Atlantic where a positive bias of up to +80 cyclones is found. Most impressively amongst all variables, this bias of the cyclone frequency is radically reduced and almost completely absent in the

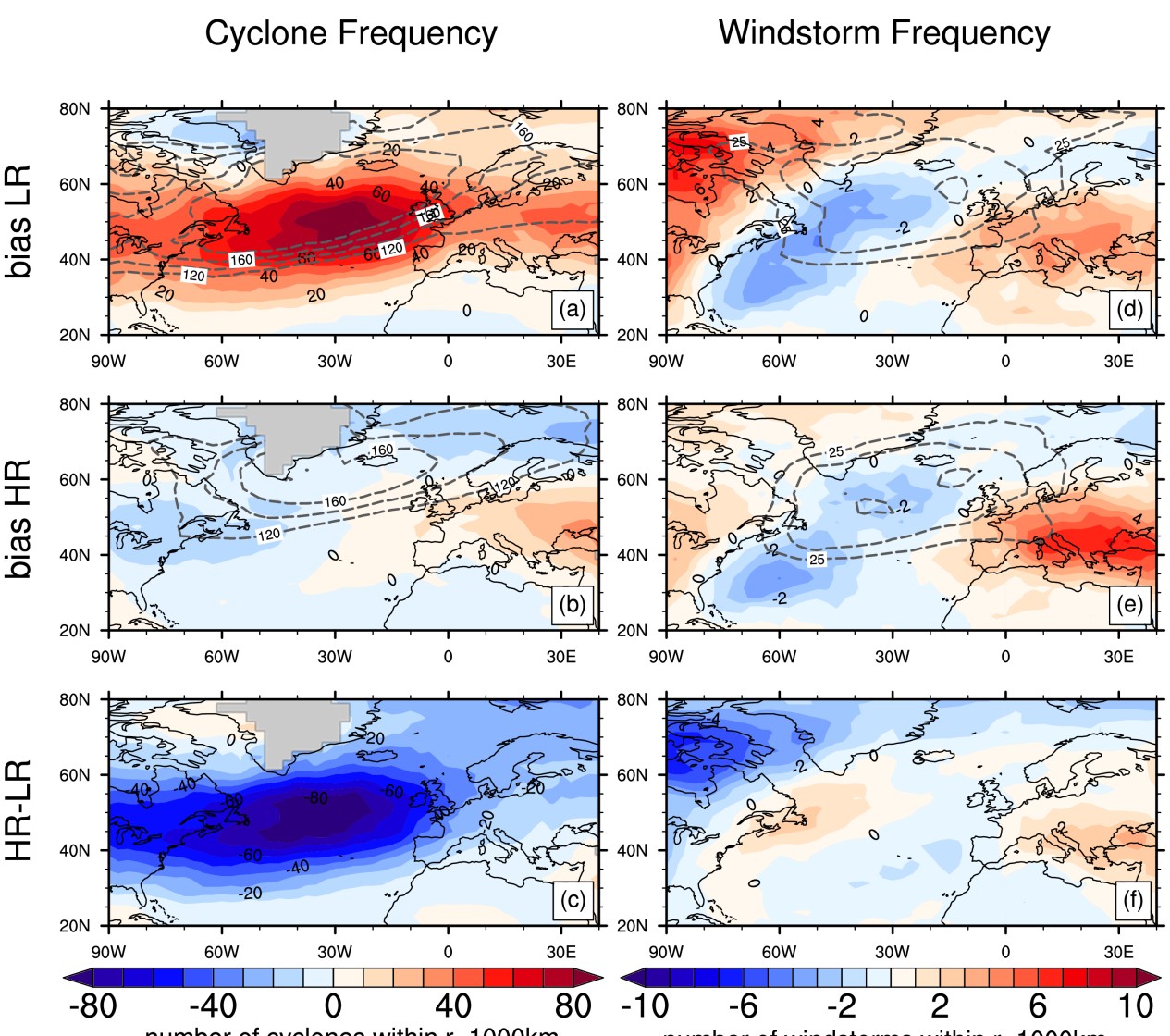

**Figure 3.** Same as Fig. 2 but for the cyclone frequency (left) and the windstorm frequency (right). The grey contours, i.e. ensemble mean climatology, have the same levels as in Fig. 1 - 120-180 by 20 cyclones for the cyclone frequency and 25-30 by 2.5 storms for the windstorm frequency.

HR system (Fig. 3b). The numbers are reduced to a bias of -10 cyclones over the western North Atlantic and +10 cyclones over Europe. The increase in horizontal and vertical resolution evidently eliminates many cyclone tracks in the MPI-ESM (Fig. 3c) over the entire North Atlantic domain and adjacent continents. This effect is further discussed in Sect.4. This results in cyclone climatologies very close to those in ERA-Interim in HR (Fig. 3b).

The windstorm frequency shows a slightly different behavior. There are too few windstorms (-3) present over the western and central North Atlantic along the North Atlantic current, and too many windstorms over the continents - +3 over Europe and +5 over North America (Fig. 3d). Given that there are too many cyclones in LR, the negative windstorm bias over the Atlantic might seem contradictory, as windstorms are a consequence of strong cyclones. However, it should be highlighted that the cyclone tracking algorithm also detects weak and moderate cyclones. Thus, the windstorms displayed in Fig. 3d-f can be considered a subset of all (weak, moderate and strong) cyclones displayed in Fig. 3a-c. This suggests that the positive cyclone frequency bias in LR is caused by weak cyclones. This is confirmed by the intensity and lifetime histograms displayed in Fig. 4, which only include cyclones that pass the central North Atlantic (50°-10° W, 40°-60°N) at some point during their lifetime. While the distributions of those North Atlantic cyclones match very well between HR and ERA-Interim, the LR prediction system overestimates weak to moderate (0.7-2.2 hPa/(degree latitude)$^2$) and short- to average-lived (1-7 days) cyclones. Those cyclones are however usually not strong enough to develop a windstorm. A similar feature was reported by Kruschke et al. (2014) for the uninitialized LR runs of the previous MPI-ESM system. Although the positive cyclone frequency bias is generally weaker for the uninitialized runs, they demonstrated that it can mainly be attributed to weak and moderate systems, by illustrating a reduced bias over the North Atlantic and Europe when only intense cyclones, i.e. the strongest 25% in terms of the Laplacian of the sea-level pressure, are considered. The negative windstorm frequency bias over the central North Atlantic is therefore not contradictory. In fact it is in line with the too zonally oriented stormtrack (Fig. 2a,b), also resulting in too many windstorms over Central Europe and the Mediterranean and too few storms over Northern Europe (Fig. 3d,e). The increase of the model resolution yields an increase of windstorm frequency over the North Atlantic current (Fig. 3e) and a remarkable reduction over the Hudson Bay. The bias over South East Europe, however, is amplified. This leaves the higher resolution system with biases of -2 along the North Atlantic current and the central North Atlantic, and +6 over South East Europe (Fig. 3f).

While the exact location and magnitude of the extra-tropical circulation features over the North Atlantic and European region exhibits deviations from the observation, overall the MPI-ESM is capable of representing those dynamical quantities. Also in Müller et al. (2018), it is noted that although bias reductions from LR to HR are modest for the multitude of diagnostics they analyzed, the dynamics of the atmosphere still benefit from the increase in resolution and make this model suitable for prediction studies. We therefore proceed to analyze the deterministic decadal prediction skill.

## 3.2 Prediction skill

The anomaly correlation between the stormtrack in the LR hindcast and ERA-Interim for the winters 2-5 after initialization is shown in Fig. 5a. Although both significant positive and negative correlations are equally valuable from a mathematical point of view, a significant negative correlation, i.e. a consistently opposite prediction of the observed climate variability is inconsistent with the physically-based model setup. We thus consider only significantly positive correlations as model prediction skill. The LR system shows skill for the stormtrack over the central North Atlantic (correlation coefficient r=0.4), as well as over Canada, the Baffin Bay and the Barents Sea. However, southwestward of the climatological stormtrack maximum, over the

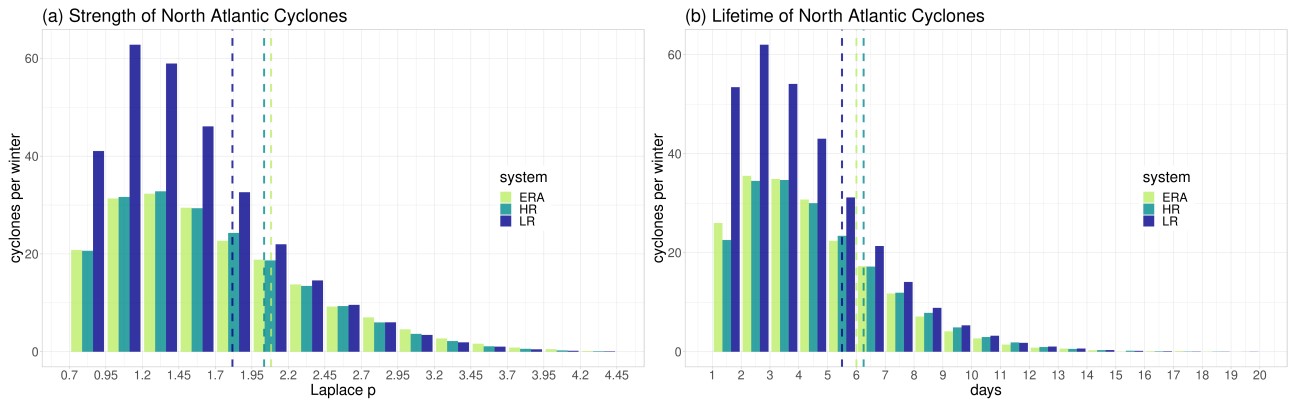

**Figure 4.** Histograms of a) the strength (max. along track Laplacian of the sea level pressure in hPa/(degree latitude)$^2$) and b) the lifetime of cyclones (in days) that pass the central North Atlantic (50°-10° W, 40°-60°N) at any time during their existence. For the hindcasts individual cyclone tracks of all 5 members and lead winters 2-5 of the initializations 1978-2012 are used, for ERA-Interim individual cyclone tracks of the period 1979/80-2016/17 are used. Vertical dashed lines denote the 0.75 quantile, i.e. the 25% strongest / longest-lived cyclones of each sample.

North Atlantic Current, where the meridional gradient of the stormtrack climatology is strongest, there is significant negative correlation (r=-0.3). This lack of skill in that area is overcome when the resolution of the dynamical model is increased. In the HR system (Fig. 5c) there is a large area of significant positive correlation over the North Atlantic Current (r=0.6) and additionally over Iceland and Central Europe (r=0.5). The improvement from LR to HR, shown in Fig. 5e, is strongest over the

5  North Atlantic Current and the tropical North Atlantic. Additionally, there are areas of statistically significant skill improvement east of the Azores, west of Iceland and over the North and Baltic Seas. Although positive anomaly correlations are not a direct consequence of bias reductions, the better representation of the average circulation and its variability does have an impact on the anomaly correlation and thus the prediction skill. Therefore, the skill improvement over those regions is in line with the average extended stormtrack in HR and the related bias reductions found on the northern side and downstream end

10  of the stormtrack. However, there is also an area of a significant reduction of the anomaly correlation for the stormtrack over Northern Canada and the Baffin Bay. Interestingly, the increase in resolution merely has an influence on the ensemble mean stormtrack bias along the North Atlantic Current (Fig. 2c), and yet appears to have a strong influence on inter-annual variability and prediction skill in that region (Fig. 5e).

15     The anomaly correlation between LR and ERA-Interim for the winter blocking frequencies is illustrated in Fig. 5b. Similar to the stormtrack, there is skill over Canada (r=0.3). Although the correlation is positive in large areas over the North Atlantic and Central Europe (r=0.2), it is only significant at a few of those grid points, e.g. south of Iceland and around the Baltic Sea. This changes in the HR system, where larger areas around and downstream of Newfoundland (r=0.4) and over Northern and Eastern Europe (r=0.3) show skill for the winters 2-5 (Fig. 5d). Also, large areas of significantly negative correlation over the

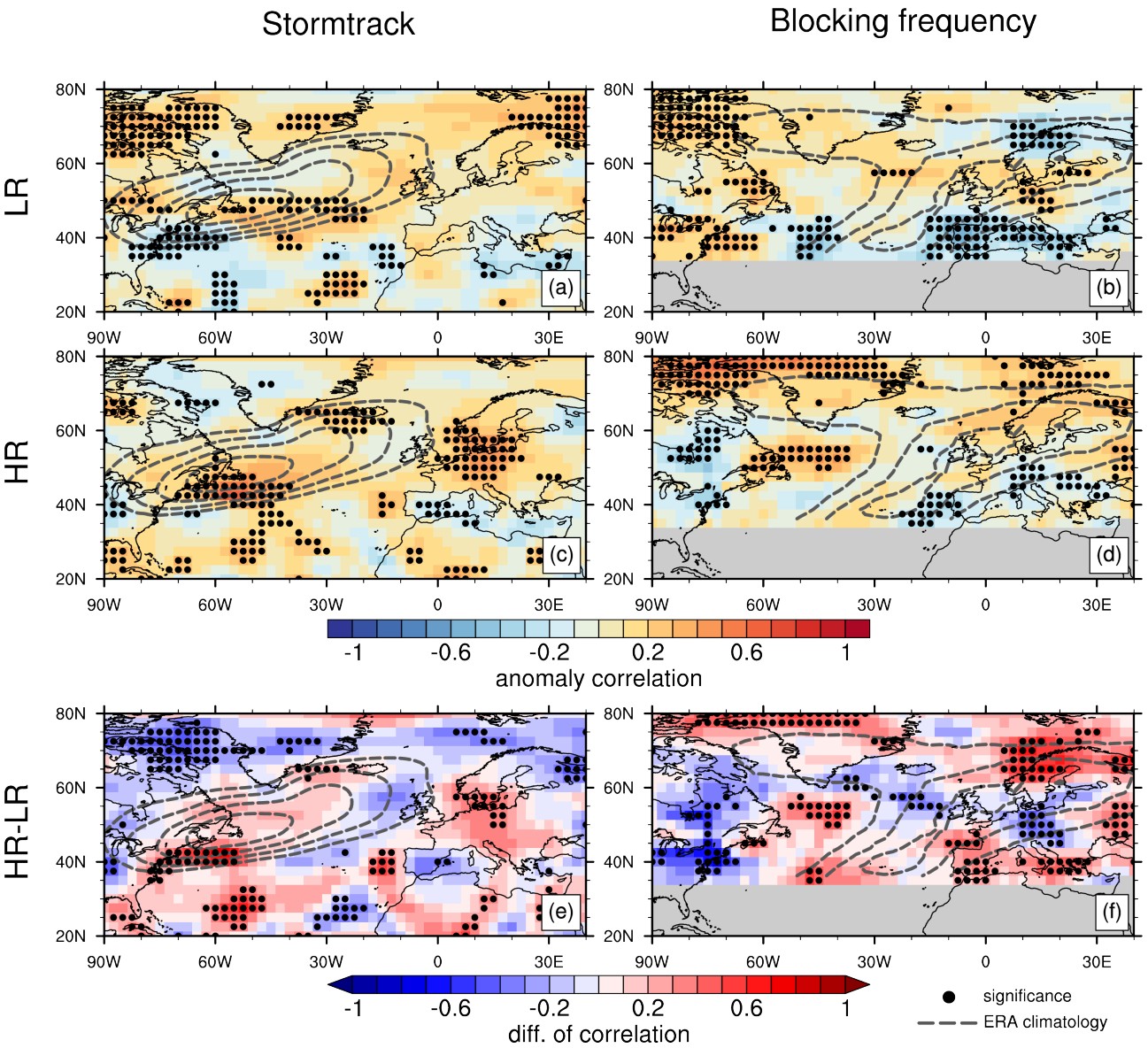

**Figure 5.** Anomaly correlation between the respective circulation quantity in ERA-Interim and LR (top row), between ERA-Interim and HR (middle row); and the difference between middle and top row (bottom row). The circulation quantities displayed are the stormtrack (left) and the blocking frequency (right). Initializations from the period 1978-2012 are used for both LR and HR and the correlation is computed for the winter (Oct-Mar) average of the hindcast winters 2-5. The dots mark significance (1000 times resampling of reanalysis-hindcast time series) at the 95% significance level. The dashed contours show the climatology of the circulation quantity in ERA-Interim (1979/80-2016/17) - as depicted in Fig. 1.

## Cyclone frequency

## Windstorm frequency

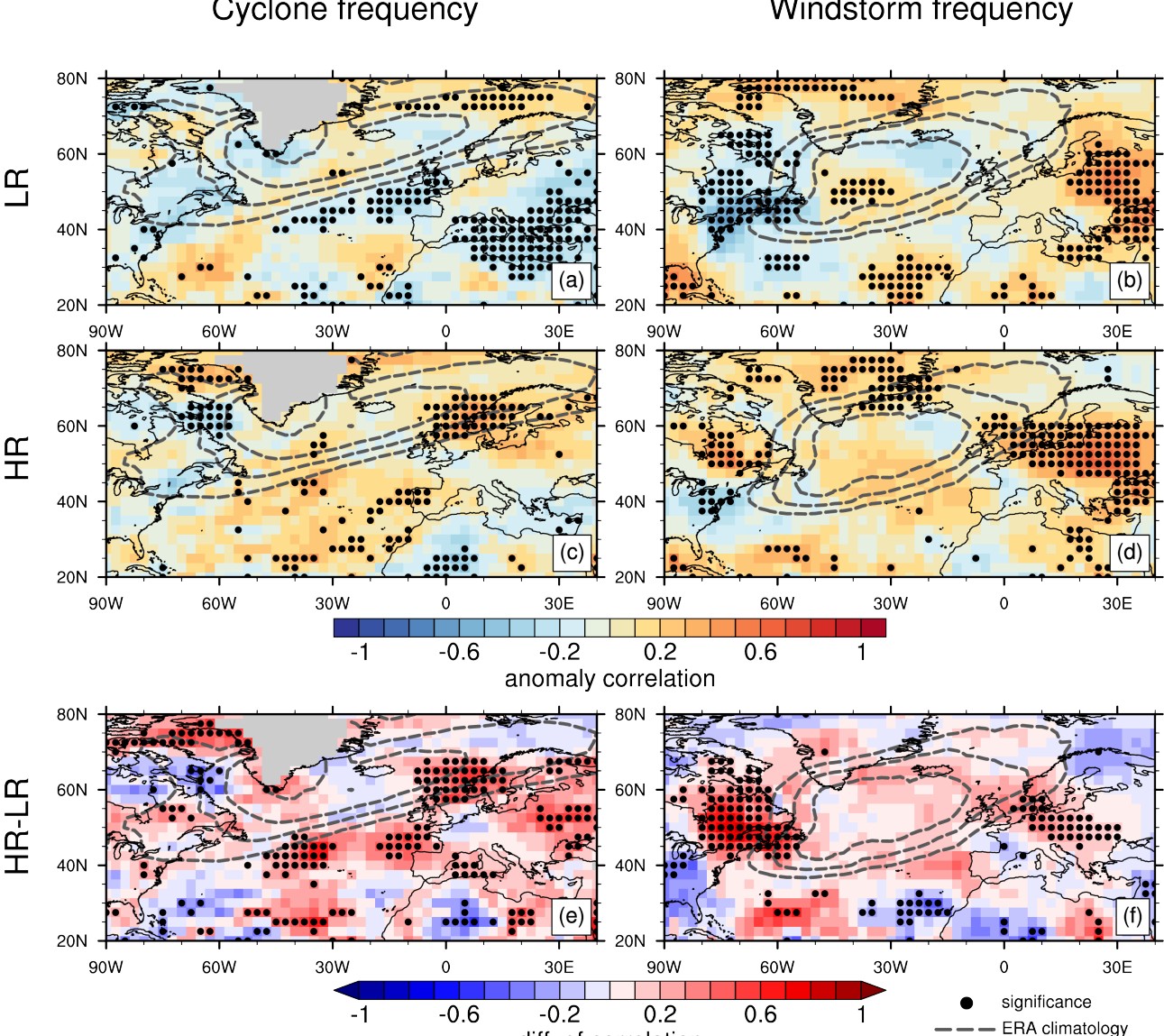

**Figure 6.** Same as Fig. 5 but for the cyclone frequency (left) and the windstorm frequency (right).

central North Atlantic around 40°N, the Mediterranean and Scandinavia, present in LR, are reduced in size or converted to positive correlation in HR. A significant improvement in correlation with respect to the blocking frequency is therefore found for several areas, such as east of Newfoundland and all around Europe, i.e. the Mediterranean, Eastern and Northern Europe (Fig. 5f) - except for Central Europe, which actually suffers from a significant decrease in correlation from LR to HR. The

5  concurrence of the skill improvement around the Mediterranean and downstream of Newfoundland and the bias reduction in

the same areas again speaks for an overall better representation of the blocking dynamics in HR.

For cyclone frequencies in the winters 2-5 in LR (Fig. 6a) there is a small area of significant skill over the Arctic Ocean north of Scandinavia (r=0.2), however the rest of the domain is dominated by even smaller or negative correlation (r=-0.3 to r=0.2). There are large regions with significantly negative correlation west of Great Britain (r=-0.3) and over the Mediterranean (r=-0.4). Once again, with the increase in resolution the skill strongly improves. In HR (Fig. 6c) positive anomaly correlation bestrides the entire North Atlantic, and the prediction is skillful (significant correlation) over a large contiguous area over the North Sea and Scandinavia (r=0.4) and at scattered grid points over the central North Atlantic (r=0.3). Only a small area over the Hudson Strait shows significant negative correlation in HR (r=0.3). Thus, the skill for extra-tropical cyclone frequencies is significantly improved through the finer resolution in large areas over the central and eastern North Atlantic, the North Sea, Scandinavia and Eastern Europe (Fig. 6e). Those areas in which the skill is improved in HR coincide with the location of the maximum bias improvement and with the more accurately represented climatological cyclone frequencies on the downstream end along the European west coast. The analysis also reveals that not only the skill for the cyclone track frequency improves, but also for the cyclone genesis frequency, i.e. the location where the cyclones form (not shown). There is significant skill improvement from LR to HR of the cyclogenesis frequency south of Greenland, over the entire eastern North Atlantic and over Northern Europe (not shown), indicating that not only the lifetime and pathway of existing maritime cyclones is improved but also the genesis of cyclones that form just off the European west coast and continental cyclones.

Prediction skill for the winter windstorms in the LR prediction system is present over the central North Atlantic (r=0.2) in the region of the maximum of the windstorm climatology and over Eastern Europe (r=0.5; Fig. 6b). A large area of significant negative anomaly correlation is located around Newfoundland (r=0.6). It is remarkable that with the finer resolution the skill increases almost throughout the entire domain, i.e. it improves over the ocean but also and most strongly over continental areas (Fig. 6f). This effect is strongest and significant around Newfoundland and over Central and Eastern Europe. This matches the results for the skill improvement of the cyclone frequencies in Fig. 6e, indicating that if the cyclone tracks are improved along the European west coast, the downstream impact of the associated wind fields of strong cyclones is also improved. Also, the skill improvement over Canada and Newfoundland again coincides with the bias reduction of the ensemble mean windstorm climatology in this region. The HR system thus produces skillful windstorm predictions over large regions of the Northern Hemisphere, e.g. eastern Canada (r=0.5), but most impressively over Central and Eastern Europe (r=0.6; Fig. 6d).

## 4 Discussion

### 4.1 Model bias

The LR system shows bias patterns that are quite common in climate models of moderate resolution: a North Atlantic storm-track which is oriented rather zonally and is southward displaced (as found e.g. for CMIP5 models in Zappa et al. 2013) and accordingly too low blocking frequencies over the eastern North Atlantic and Northern Europe (as documented e.g. by Scaife

et al. 2011). With respect to the stormtrack, the too zonal pattern in LR is to some extent corrected in HR, especially over Northern Europe. This corresponds to findings from Müller et al. (2018), who note a reduced bias of the atmospheric jet stream position in the northern extra-tropics, a decrease of the storm track bias over the northern North Atlantic and an increased storm activity over northern Europe for the uninitialized runs of the higher resolution system. Thus, in HR the stormtrack reaches fur-

ther across the North Atlantic and is more tilted towards Northern Europe and to that effect is closer to the observed stormtrack than in LR. This is conform with the results presented by Zappa et al. (2013) who state that only the higher resolution models of their study are able to capture the tilt of the winter stormtrack. However, the general bias of a southward displacement over the central North Atlantic is still apparent in HR, in agreement with the too zonal North Atlantic Current identified in both model versions by Müller et al. (2018).

The corresponding deficit in blocking over the Atlantic, which is strongest near the end of the stormtrack in both systems, has also been reported in previous works (e.g. by Scaife et al., 2010; Davini et al., 2017). While other studies that investigate the effect of the resolution on the blocking frequency bias find an increase over central and eastern Europe (Berckmans et al., 2013) or northern Europe (Davini et al., 2017), HR shows the increase of blocking frequency over the central and eastern North Atlantic - as seen in Müller et al. (2018). However the changes in our study are small and a widespread negative bias remains

along the European coast. A comparable result is documented in Jiang et al. (2019), who state that the underestimation of blocking frequencies over the North Atlantic found in their lower resolution (1°) model version mainly persists in the higher resolution (0.25°) version. They conclude that, in contrast to North Pacific blocking, North Atlantic blocking is mainly driven by low-frequency eddies, which are not influenced by the higher resolution.

In contrast to the previous discussed quantities, cyclone frequencies are affected very strongly by the increase in resolution.

The intense positive cyclone frequency bias, which is visible over the entire North Atlantic in LR, is almost entirely removed in HR. A similar bias tendency and pattern, however not as strong, is reported in previous studies with MPI-ESM predecessors (Kruschke et al., 2014; Bengtsson et al., 2006). First analyses reveal that the strong bias in our study appears to be the result of a combined effect of the LR system and the initialization (see supplement Fig. S1), as there is no such bias present in the respective uninitialized simulations of the LR system (Fig. S1) nor in the initialized HR system (Fig. 3b). In fact, we find that

this strong cyclone frequency bias is, in the same order of magnitude, already inherent in the initialized runs of the previous MPI-ESM-LR version termed Baseline1, analyzed by Kruschke et al. (2014) (Fig. S2) - however, they only show biases for the uninitialized simulations which do not exhibit this cyclone frequency bias (Fig. S1, S3). The contributions to the bias through a) the different reanalysis used in Kruschke et al. 2014 (Fig. S3) and b) the different model physics of the meanwhile advanced prediction system (Fig. S4) are negligible. The detection and a detailed analysis of the factors and processes provoking this

bias in the LR system are beyond the scope of this study and remain to be addressed in future investigations. Nevertheless, we would like to emphasize that there is no such cyclone frequency bias in the HR system, which makes it suitable for studies on the variability and decadal prediction skill of extra-tropical cyclones.

In line with the southward shifted exit region of the stormtrack in LR there is a positive windstorm frequency bias over Central Europe and the Mediterranean. Also, the central North Atlantic is experiencing an underestimation of windstorms. A

similar bias pattern is identified by Kruschke et al. (2016) for the uninitialized runs of a previous MPI-ESM system and by

Befort et al. (2019) for the ECMWF's seasonal forecast systems S3 and S4. This bias is not corrected by the higher resolution, neither in our study nor in the higher resolved S4 system in Befort et al. (2019), and the bias over the Mediterranean in HR is even aggravated. On the other hand, a slight bias improvement is found along the North Atlantic current and a strong improvement over Canada. A better representation of near surface processes would seem to be a likely cause, as the windstorms are identified from low level wind speed, but neither sea surface or 2m temperatures nor sea ice fraction show a considerable improvement over that area in the higher resolution system (Müller et al., 2018). Only the sea-level pressure shows a bias reduction over Canada, however the opposite is the case over the North Atlantic Current.

Although the bias and the anomaly correlation are per se unrelated, they are both important metrics to assess the model's performance to correctly represent the mean state and the variability of the atmospheric circulation. If they appear to be improved in the same location this does not imply a causal interrelation. However, it all the more indicates that local physical processes are improved in the higher resolution prediction system.

## 4.2 Prediction skill

With respect to the decadal prediction skill, for the stormtrack a strong (statistically significant) skill improvement is found along the North Atlantic Current and over Central Europe. Given the important role of surface heat fluxes and local SST gradients for the dynamics of the stormtrack (Brayshaw et al., 2011), these are likely sources of improved atmospheric variability in the HR system. The skill improvement over Central Europe is in line with the bias improvement of the stormtrack at its downstream end (stronger tilt and downstream extension) shown in our study as well as with reduced sea surface temperature and salinity biases over the eastern and northern North Atlantic found in Müller et al. (2018) for the uninitialized runs of the same model system. However, the influence of local SST gradients on the stormtrack skill improvement along the North Atlantic current is debatable, given the mostly unchanged bias of the North Atlantic Current in HR documented in Müller et al. (2018).

The skill for blocking frequencies over the North Atlantic and European domain is basically non-existent in the LR system, it shows only skill over Canada. A similar pattern of skill is found in Athanasiadis et al. (2014) for seasonal forecasts performed with the CCMC model, which is as well based on the ECHAM. They further find that this lower resolution model (CCMC; atm. ~1.875°) underrepresents the variance of blocking in the eastern North Atlantic more than the higher resolution model (UKMO; atm. 0.83° x 0.55°) of their study. We find in HR a significant improvement in the anomaly correlation of the blocking frequency downstream of where the stormtrack skill is improved, i.e. over the central North Atlantic and Northern Europe and the Mediterranean. The latter coincides with a strong bias reduction in that area. This matches with the results of Athanasiadis et al. (2014), where as well only the higher resolution system shows skill for blocking frequencies in the Euro-Atlantic sector - in their case over the eastern North Atlantic and Central Europe. They state that those are preliminary the regions where blocking activity is strong and related to the NAO variability. If this relation was as well valid for our simulations, the skill improvement for blocking over Northern Europe and the Mediterranean would be in line with NAO amplitudes reaching further towards Europe in HR than in LR, as found by (Müller et al., 2018).

The strong misrepresentation of the cyclone frequencies in LR results in no decadal forecast skill throughout the North Atlantic and European domain. But with the increase in resolution not only the striking climatological bias reduction is achieved, but also the prediction skill is improved throughout the entire domain - significantly over the central and eastern North Atlantic and Northern Europe - resulting in skillful cyclone frequency predictions over Northern Europe in HR. The improved representation of cyclones in this region may also be beneficial for the prediction of blocking over Scandinavia (where the skill in HR is significantly improved), as cyclones can contribute to downstream blocking formation through eddy vorticity forcing (Shutts, 1983) and diabatic processes (Pfahl et al., 2015). Apart from the removed initialization effect in HR, a more accurate representation of smaller-scale diabatic processes may be a reason for the increased forecast skill of cyclones at the southern flank of the main stormtrack, over the subtropical North Atlantic and the Mediterranean, as moist processes are thought to be particularly important for such subtropical systems (e.g. Davis, 2010). The fact, that Kruschke et al. (2014) find the decadal forecast skill to be higher for strong cyclones than for all cyclones in the former LR system could as well be related to the initialization effect apparent in LR. As strong cyclones are not affected by this initialization induced bias, likely their prediction skill is as well more credible.

Although the forecast skill for 10m wind speeds and wind energy output only differs slightly between different ocean initializations (Moemken et al., 2016), this study reveals that an increased model resolution has a large impact on the hindcast skill of synoptic scale features, such as cyclones and windstorms. Thus, in line with these skill improvements in the cyclone frequency, the skill for windstorms also improves significantly over North-East and Central Europe, i.e. south of the cyclones' signal. This matches with the general south-eastward displacement of the maximum wind speeds relative to the cyclone center (Leckebusch et al., 2008). Müller et al. (2018) deduced from the stormtrack bias changes they found in the uninitialized higher resolution runs that more storms entering the northern European region can be expected, relative to LR. Although this cannot be confirmed with respect to the windstorm frequency climatology in our study, its prediction skill is significantly improved from the North Sea through Eastern Europe. Different studies suggest that a better representation of the North Atlantic current in the model would contribute to a better representation of the storm track (e.g. Brayshaw et al., 2011; Scaife et al., 2011) and thus would probably lead to increased downstream predictive skill for cyclones and windstorms (Kruschke et al., 2014). Although the improvement for the North Atlantic current in terms of sea surface temperature and salinity is small for the HR system, as reported by Müller et al. (2018) for the uninitialized runs, this study found improved decadal prediction skill downstream of the stormtrack, not only for cyclones and windstorms but also for blocking frequencies. This indicates that the variability of strong North Atlantic cyclones causing windstorms in North and Central Europe is much better captured by HR.

## 5 Conclusions

This study evaluated the response of the deterministic decadal forecast skill of the atmospheric extra-tropical winter circulation to an increase in the resolution of the forecast system. This was performed under otherwise unchanged conditions, i.e. the same numerical model, initialization technique and parametrization were used and only the resolution of the model was changed.

The two hindcast sets (LR: atm. ∼1.8°, ocean ∼1.5° and HR: atm. ∼0.9°, ocean ∼0.4°) were initialized in the period 1978-2012 and evaluated for the winters 2-5 after the initialization, using 5 members each. Those hindcasts were performed with the MiKlip pre-operational decadal prediction system, based on the MPI-ESM. The forecast skill was analyzed over the North Atlantic region in terms of anomaly correlation for the stormtrack, blocking frequency, cyclone frequency and windstorm

frequency. ERA-Interim, i.e. the winterly averages of the four quantities between 1979/80 and 2016/17, served as the reference dataset. The analysis of the ensemble mean model bias has provided additional insights into the modified atmospheric dynamics and into possible sources of improved forecast skill in the higher resolved system.

In summary we demonstrated an improvement of the mid-latitude dynamics in the North Atlantic region with an increase in the model resolution. This comprises an improvement of both the mean state (climatology) and the temporal variability (decadal

prediction skill) for the different extra-tropical circulation metrics. Although there are yet no other studies on this topic with respect to decadal time scales, our results are in agreement with findings from seasonal prediction studies (e.g. Prodhomme et al., 2016; Befort et al., 2019), who showed skill improvements for blocking, windstorm and cyclone frequencies when the same model is used and only the resolution is increased.

The improvements found in our study for the different metrics follow a physically consistent line of argument and the

areas of improved forecast skill are crucial regions for the genesis and intensification of synoptic weather systems over the North Atlantic and for their impact on Europe. Thus, we identified a significant improvement of the stormtrack skill along the North Atlantic Current (i.e. the source region of synoptic eddies), a downstream improvement of the cyclone frequency skill over the central North Atlantic (where the synoptic systems intensify), and finally improved skill of the cyclone, windstorm and blocking frequencies over the European continent (i.e. the impact area). Additionally, not only does the prediction skill

improve with a finer resolution (HR vs. LR), the HR system itself offers significant deterministic decadal forecast skill for the extra-tropical circulation metrics in large regions over the North Atlantic and Europe (HR vs. ERA-Interim) for the considered lead time of 2-5 winters.

By analyzing different but physically linked extra-tropical circulation metrics, this study contributes to the elucidation of the processes that lead to the decadal prediction skill in the North Atlantic region. Our results are encouraging as they docu-

ment the successful advancement of decadal prediction systems and in particular of the deterministic decadal prediction skill of extra-tropical features and extreme events. However, future studies using different prediction systems, possibly of higher resolution and larger ensemble sizes, and especially rather process oriented analyses will be needed to shed further light on the robustness of our results and the sources of the presented skill.

*Competing interests.* No competing interests are present.

*Acknowledgements.* The authors would like to acknowledge funding from CoreLogic SARL Paris and from the Federal Ministry of Education and Research in Germany (BMBF) through the research program MiKlip II (FKZ: 01LP1519A, 01LP1519B, 01LP1520A) and the CMIP6-DICAD project (FKZ: 01LP1605D). We acknowledge support by the Open Access Publication Fund of the Freie Universität Berlin. We would also like to thank the anonymous reviewers for their valuable comments.

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
