# Peer review of "Improvement in the decadal prediction skill of the North Atlantic extra-tropical winter circulation through increased model resolution"

_Earth System Dynamics, 2019_

## Referee Comment (RC1) · Anonymous Referee #1 · 30 May 2019

This study explores the prediction skills of MiKlip decadal hindcast system with low and high resolution for stormtracks, blocking, cyclone and windstorm frequencies. The model biases and prediction skills are assessed by comparing the synoptic variability in the model outputs for lead years 2-5 with ERA-interim.

For the most part, I am satisfied with the data, methods, and interpretations offered, however, writing needs to be improved. Though the sentences are grammatically correct, they often sound awkward.

[Figure]

Specific comments:

p1, l.20: the first sentences sound as if extra-tropical circulation is important because it may be linked to extreme events. Isn't it important in a more general sense? After all, it is not a paper on extremes. Will be good to discuss the motivation in a broader context.

The term 'stormtrack' is confusing when used along with the cyclone frequencies - they are sometimes used interchangeably (not in this paper). Though the Methods describe what is meant by the stormtrack, I recommend commenting on the difference early in the manuscript (maybe even in the abstract)

The same goes to lead years/winters - it is worthwhile explaining which months are considered. I only found this information in Figure captions.

p2, l28: comment on what parametric bias adjustment approach is.

p10,l1: The word 'shift' often implies change in time, consider revising

p10,l13: I would be more precise here and stick to the words used in the Methodology, i.e. 'open' and 'closed'. Otherwise, you need to clarify what you mean by weak/strong cyclones.

p.10,l14: I would like to see a figure confirming that positive bias is due to the weak and/or short leaved cyclones. p.10,l.15-16: How do you explain then negative windstorm vs positive stormtrack anomaly over the Atlantic?

p10, l.31: I can see a discussion on negative correlations further in this section (e.g.p13, l7)

p12,l6: I How about a strong reduction of skill over Northern Canada and the Barents Sea?

p.12, l3: 'significant skill improvement' - the authors probably mean that HR model shows statistically significant correlation with ERA-Interim at some points. In my opin-

[Figure]

ion, though, this statement makes an impression that skills of model prediction have become really good (so say at least 'statistically significant skill improvement' or re-phrase). More important, the prediction skills, as shown in the paper, are remarkably low for most part of the region, but this message is not conveyed by the paper - will be good to see more discussion on that.

Figures 4-5: In line with the previous comment, it will be interesting to calculate the percentage of area that is significantly (positively?) correlated with ERA-Interim. This number can be added to each subplot.

Discussion and Conclusions: this section is too long, consider shortening. Parts of the discussion may be moved to the Results. The last sentence of the article is not clear, please revise.

Fig4: significant at what level?

Minor comments:

p2,l.11: remove comma before dash. p2, l18-19: put references in brackets p2, l29: did you mean more skilfull ? Skilful is misspelled. p5, l11: 1000 time - should '1000 time steps' be better? p10, l6: should read 'these results'

---

## Referee Comment (RC2) · Anonymous Referee #2 · 3 Jun 2019

General comment:

The presented manuscript by Schuster et al. addresses the scientific relevant question as to how spatial resolution of a specific decadal prediction system influences the forecast quality of North Atlantic circulation measures. It is well within the scope of ESD and will find interest from readers in the research areas of decadal prediction, forecast quality assessment, climate modeling, and extra-tropical atmospheric dynamics. The study is well structured and offers novel results regarding the effect of spatial resolution in a decadal prediction framework for measures relevant to society and economy.

[Figure]

There are however some shortcomings in the argumentation supporting the conclusion likely due to an incomplete or imprecise description of the applied methods. The analyses provide insufficient explanation for some of the key results. The clarity of the manuscript will benefit from a revision of the writing.

Recommendation:

Re-Submission after few additional analyses and clarification of some results and applied methods

Specific comments:

1) The applied methods are often not clear. The use of an "evaluation software" is mentioned (P5L3). What does it actually do? When is the ensemble mean calculated, e.g. are the shown correlation maps means of correlations or correlations between ensemble mean and reference. Please provide clarification and add the applied calculation methods/equations. Could be as appendix/supplement.

2) The study suggests a direct relation between the mean bias of the ensemble mean and the anomaly correlation of the ensemble mean to the reference for one and the same diagnostic. The correlation is insensitive to the mean bias on grid cell level, hence anomaly. It appears large parts of the result section and conclusions are based on the assumption that a reduction of mean error/bias leads to higher anomaly correlations for the same analyzed quantity. This has to be revised substantially.

3) The hindcasts are presumably not post processed, e.g. corrected for time-varying bias, trend-adjusted, etc? Please clarify and state why this might be not necessary. Why is the approach of correcting biases of this study different to Kruschke et al?

4) Spatial resolution has been discussed to be a serious limiting factor to correctly reproduce climate mean state and variability in the context of decadal prediction (e.g. Hewitt et al. BAMS 2017, Smith et al. QJRMS 2016). This should be mentioned more prominently in the motivation and put in context of this study in the discussion.

There are numerous studies about the effect of resolution in climate models in general including the effect on North Atlantic circulation measures (e.g. Davini et al. J ADV MODEL EARTH SY, 2017). How do they compare to this study?

5) The difference in mean bias for LR and HR in cyclone frequency is striking and given the very small differences in stormtrack activity somewhat unexpected, e.g. at 30W, 50N (Fig 2a vs Fig 3a). The result is apparently similar to Kruschke et al 2014. Kruschke et al compare uninitialized experiments in LR to NOAA's 20th Century Reanalysis. In their study the mean bias is up to 25 systems per winter over the North Atlantic and they mention a possible underestimation of cyclone frequency of the reanalysis. This seems at odds with what is shown here: A mean bias of up to 80 systems and more per winter in comparison to a different reanalysis product. Please discuss this Is it possible to estimate how much is due to the applied tracking method? One suggestion could be to interpolate the HR hindcast to the lower resolution and repeat the analysis. Will that change the results? This could be done for a single member and put as appendix. It is mentioned that LR overestimates weak and moderate systems. Why?

6) The ensemble spread is unfortunately not used or shown for any of the analyses. How is the spread different between LR and HR? Is the reanalysis within the spread?

7) When analyzing absolute numbers (here for blocking, cyclones and windstorms) ties have to be considered in the correlation calculation, ie seasons with the same number of events. Presumably ties are not taken into account as the manuscript does not mention it. 2 possible solutions: i) mask regions with a large number of ties ii) use a different correlation coefficient, e.g. Kendall's Tau B. Otherwise the correlation value could be misleading and statistical significance becomes meaningless, especially in regions with few events per season. There is a significant negative correlation in windstorm frequency in LR over Eastern Canada and a significant positive correlation in HR over the same region. This could be an example of too many ties.
8) Related to the above point: Cyclone frequency is apparently masked in regions with high orography. This can be seen in Fig 5. Why is there no mask in Fig 3? What about wind storms. Why are windstorms not masked? Please also consider masking regions with few events per season. There is a mask for blocking. Please state why.

9) The discussion is not critical enough. The reader gets the overall impression of a nearly perfect prediction system regarding the analyzed quantities. Mentioning the correlation value could sometimes already be enough to put the results in perspective. There are some inconsistencies as mentioned above that should be discussed. There is only one sentence P16, L15ff with reference to previous studies with similar objectives. Please add some references or state the lack thereof. See also point 4)

Minor comments:

i) The title suggests an analysis of the entire NH. Please correct. Consider adding the word "deterministic" in the title

ii) "Anomaly correlation" and "skill" are used as synonyms throughout the manuscript. Please state that deterministic skill is assessed through anomaly correlation somewhere in the paper and in the abstract.

iii) There is no reference for the "common shortcoming of climate models" of a too zonal stormtrack in the introduction.

iv) P1L1: The acronym MiKlip is not explained

v) P1L8: "functional chain" is not clear

vi) P1L11ff: Newfoundland is not "downstream" of the stormtrack.

vii) P1L20ff: Please add reference for this paragraph

viii) P2L8: "sectors" is most likely the wrong word

ix) P2L20ff: restructure sentence: "One result..."

x) P3L1: specify "lower resolution"

xi) P3L8: "functional chain?"

xii) P3L22: change "variables" to "diagnostics" or similar. Variable is not the correct word.

xiii) P3L29: same, please check the entire manuscript

xiv) P4L12ff: "However...": Please rephrase

xv) P5L1: add "deterministic". See points i) and ii)

xvi) P3L2: centered or uncentered anomaly correlation? See point 1)

xvii) P6L32ff: This is unclear and probably wrong somehow. What kind of percentile is used? Is it the same one in LR and HR? This might explain why the difference in cyclone frequency is not apparent in windstorms

xviii) P7L2ff: change "nicely illustrated"

xix) P7L31: the value in brackets is easily misunderstood. Maybe: -3% of a total of X% days in one season

xx) P10L18: change "implying" to "could be due to" or similar

xxi) P12L4ff: see point 2 for the whole paragraph

xxii) P12L22ff: see point 2

xxiii) P13L35ff: improvement in cyclone frequency improves windstorm frequency? Specifically along the European western coast? P10L12ff highlights the differences of the 2 diagnostics

xxiv) P15L8: Muller et al 2018 show a decrease of MSLP bias in the Eastern North Atlantic but an increase in the Western North Atlantic in HR. It is therefore only partially "in line".

xxv) P15L25ff: see point 2, for blocking + cyclones

xxvi) P5L10: Please provide a reference or calculation method for the statistical significance. Is the calculation method different between correlation significance and significance for the differences in correlation?

---

## Author Comment (AC1) · 10 Jul 2019

Response to Comments of Anonymous Referee 1 [R1]

- R1 – comment 1:
  p1, l.20: the first sentences sound as if extra-tropical circulation is important because it may be linked to extreme events. Isn't it important in a more general sense? After all, it is not a paper on extremes. Will be good to discuss the motivation in a broader context

[Figure]

**Response to R1 – comment 1**:
The paper is partly on extremes, as windstorms are identified by the exceedance of the local 98th percentile of the surface wind and blocking is identified by blocked flow for min. 4 consecutive days – these are extreme events by definition. However, the extratropical circulation is indeed important in a more general sense - thank you for the remark. The original text was changed to: "The extra-tropical circulation plays an important role for the redistribution of energy in the atmosphere. The prevailing westerlies and the embedded cyclones and anticyclones determine the weather and climate of the mid-latitudes, assisting in balancing temperature and humidity contrasts between the tropics and polar regions. Natural climate variability as well as externally forced climate change determine fluctuations in the circulation and thus i.a. the frequency of extremes such as strong cyclones, intense windstorms or phases of blocked flow. The consequences of such features include extremes in temperature, precipitation/drought and wind speed, often accompanied by immense damage and harm (e.g. Leckebusch2004, Ulbrich2009, Sillmann2009, Pfahl2012, DeutscheRueck2018)."

• R1 – comment 2:
The term 'stormtrack' is confusing when used along with the cyclone frequencies – they are sometimes used interchangeably (not in this paper). Though the Methods describe what is meant by the stormtrack, I recommend commenting on the difference early in the manuscript (maybe even in the abstract)
**Response to R1 – comment 2**:
p.1, l.3 was changed to: "Four metrics - the Eulerian stormtrack, and Lagrangian blocking, cyclone and windstorm frequencies - are analyzed . . . ". We believe that this phrase makes it clear that we differ between the two metrics. The reader can learn about the details of the methodologies in section 2.2.

• R1 – comment 3:
The same goes to lead years/winters - it is worthwhile explaining which months

are considered. I only found this information in Figure captions

**Response to R1 – comment 3**:

The fact that we are analyzing the winter half year (Oct-Mar) is e.g. stated at p.4, l.6: "We will therefore focus on the winter circulation and evaluate averages of the stormtrack and blocking, cyclone and windstorm frequencies from October through March."

However, to comply with both reviewers' requests for more specific information on the evaluation procedure, the entire paragraph was revised to be more precise and now reads (p.5, l.1ff):

"To derive the deterministic skill of the two forecast systems, we focus on the temporal variability and analyze the anomaly correlation for the winters 2-5 (Oct-Mar), following the Decadal Climate Prediction Project (DCPP, Boer2016) protocol. That means that we calculate lead time dependent anomalies of the circulation measures. This is a simple and robust approach to account for a possible lead time dependent mean bias, i.e. drift. For each of the initialization experiments (1978, 1979, ...) the ensemble average (5 members) of the temporal mean of the 4 contained lead winters is calculated per grid point. This forms a new ensemble mean time series of the lead winters 2-5. This time series serves to calculate the climatology (temporal mean) and to calculate the respective anomaly time series. The time series of those anomalies of the hindcasts is then correlated (Pearson) to the time series of anomalies of the reanalysis. In decadal prediction studies, this procedure is usually repeated for each lead time, thus lead year 1, lead year 2-5, lead year 6-9 - it is therefore referred to as lead time dependent anomaly correlation. In our study we only show results for one lead time: lead winters 2-5. The initialization of the hindcasts takes place in October, this means the first full winter that we analyze is the second winter, i.e. the months 12-17 (Oct-Mar) after initialization. This evaluation procedure is part of the decadal climate prediction evaluation software that was designed within the MiKlip project (Illing2014) and is applied for this study."

[Figure]

- R1 – comment 4:
  p2, l28: comment on what parametric bias adjustment approach is.
  **Response to R1 – comment 4**:
  The wording parametric or non-parametric corresponds to the way how to adjust lead time dependent bias (drift). On the one hand, it is feasible to assume a lead time dependent bias and to fit a curve. Kruschke et al. (2016) called this approach parametric. On the other hand, DCPP recommends to calculate lead time dependent anomalies for each lead year separately (Boer et al., 2016). This is a non-parametric approach. We used the DCPP recommendation in our manuscript.

- R1 – comment 5:
  p10,l1: The word 'shift' often implies change in time, consider revising
  **Response to R1 – comment 5**:
  We think that the structure of the sentence makes it clear, that a spatial shift in LR compared to the reanalysis is meant.

- R1 – comment 6:
  p10,l13: I would be more precise here and stick to the words used in the Methodology, i.e. 'open' and 'closed'. Otherwise, you need to clarify what you mean by weak/strong cyclones.
  **Response to R1 – comment 6**:
  The following sentence was added to p.6, l.20: "Only cyclones that lived for more than 24 hours and reached a Laplace larger than 0.7hPa/(degree latitude)$^2$ at least once during their lifetime are selected for evaluation."
  p.10 l.13 was changed to:" However, it should be highlighted that the cyclone tracking algorithm also detects cyclones in their weak phase, as long as they reach the 0.7hPa/(degree latitude)$^2$ threshold once during their lifetime." This means, that a cyclone can be part of the evaluation, which lives a couple of days but is generally weak in terms of its Laplacian of the pressure (< 0.7hPa/(degree

latitude)$^2$), but it reached the intensity criterion for exactly one time step and therefore was included in the evaluation.

- R1 – comment 7:
p.10,l14: I would like to see a figure confirming that positive bias is due to the weak and/or short leaved cyclones. p.10,l.15-16: How do you explain then negative windstorm vs positive stormtrack anomaly over the Atlantic?
**Response to R1 – comment 7**:
To answer this question we selected and evaluated cyclones that, at any time during their lifetime, pass through the central North Atlantic (50-10°W,40°-60°N) - the region where the bias in Fig. 3a is strongest. This analysis is performed for individual cyclone tracks of all initialization experiments between 1960-2012, all 9 forecast winters and all 5 members, for LR and HR respectively.
The intensity histogram (review response Fig. 1) of cyclones that pass the central North Atlantic shows that weak cyclones are more numerous in LR than in HR or ERA-Interim. Although LR overestimates the frequency of weak cyclones in that region, the frequency of the strong cyclones, in that case the strongest 5% of cyclones, i.e. bars to the right of the dashed line, is reproduced quite well in LR. This threshold (dashed lines) in HR (2.87 hPa/(deg.lat.)$^2$) is closer to ERA-Interim (2.98 hPa/(deg.lat.)2) than LR (2.65 hPa/(deg.lat.)$^2$) is to ERA-Interim - but mainly due to the generally larger number of events in LR. Overall, the shape of the intensity distribution for cyclones passing that region is much more similar, and actually almost identical, between HR and ERA-Interim, than between LR and ERA-Interim. The lifetime histogram (review response Fig. 2) of cyclones that pass the North Atlantic region also shows that short-lived cyclones in this region are more frequent in LR than in HR or ERA-Interim. Again the shape of the distribution matches very well between ERA-Interim and HR.
Regarding the second part of the comment: The stormtrack is calculated from the variance of the geopotential height in the synoptic band. Within this quantity,

there are many systems included which do not produce windstorms, as this is the variability of all geopotential values (high pressure as well as low pressure systems, and strong ones as well as weak ones). Only strong low pressure systems can be related to windstorms. It cannot be expected that the signals of total variance of geopotential height (stormtrack) are identical to those of the windstorms since the distribution of strong and weak cyclones changes differently as discussed above.

• R1 – comment 8:
p10, l.31: I can see a discussion on negative correlations further in this section (e.g.p13, l7)
**Response to R1 – comment 8**:
As we lined out in the paper, it is not desirable to have a deterministic prediction model to continuously predict the opposite of the observed quantity. We therefore stick to the opinion that negative correlations should not be considered skillful and will therefore not discuss them in detail. If anything, then the message of our paper is that the amount of negative correlations is reduced in HR. This is covered by the discussion of positive differences between HR and LR (Fig. 4e, 4f, 5e, 5f).

• R1 – comment 9:
p12,l6: I How about a strong reduction of skill over Northern Canada and the Barents Sea
**Response to R1 – comment 9**:
The following sentence was added to p.12, l.6: "However, there is also an area of significant reduction of the anomaly correlation for the stormtrack over Northern Canada and the Baffin Bay."

• R1 – comment 10:
p.12, l3: 'significant skill improvement' - the authors probably mean that HR model

shows statistically significant correlation with ERA-Interim at some points. In my opinion, though, this statement makes an impression that skills of model prediction have become really good (so say at least 'statistically significant skill improvement' or rephrase). More important, the prediction skills, as shown in the paper, are remarkably low for most part of the region, but this message is not conveyed by the paper - will be good to see more discussion on that.

**Response to R1 – comment 10**:

Referring to Fig. 4.e in this line (p.12 l.3), we are not discussing the skill of HR compared to the reanalysis but rather the change in skill from LR to HR. Thus, we indeed mean that the change from LR to HR shows an improvement, e.g. from low or (significant) negative correlations (no skill) in LR to significant positive correlations (skill) in HR, i.e. an improvement in skill or one could also say a statistically significant improvement in anomaly correlation.

Please note that deterministic decadal prediction skill in terms of anomaly correlation is generally low for model variables other then surface temperature (compare skill of precipitation in Kadow et al., 2016 or cyclone frequency in Kadow et al. 2017 to skill of surface temperature in Pohlmann et al., 2013).

We changed the original wording to: "...statistically significant skill improvement..." and added values of correlation in the text to put results into perspective.

- R1 – comment 11:

Figures 4-5: In line with the previous comment, it will be interesting to calculate the percentage of area that is significantly (positively?) correlated with ERA-Interim. This number can be added to each subplot.

**Response to R1 – comment 11**:

As the climatologies of the circulation quantities show, the values of the anomaly-correlation are not equally important anywhere in the displayed domain. A significant positive correlation e.g. along the maximum blocking frequency is more relevant than a significant positive correlation over a region with very low frequencies. Therefore, we think the suggested percentage of grid points with significant positive correlation would be misleading, as it would weigh "irrelevant" regions equally to "relevant" regions. We produced the figure (review response Fig. 3) to answer the reviewer's question but will not show it in the paper for the stated reasons. The figure shows that for all circulation quantities the number of grid points (in the North Atlantic domain) with significant negative anomaly correlation is reduced in the higher resolution system and the number of grid points with significant positive anomaly correlation is increased in the higher resolution system - supporting the theory of improved physical processes throughout the region.

We understand the referee's main point with this comment is, similar to R1 comment 10, that we emphasize the positive effects of the resolution and the reader could think we suggest that HR is the perfect decadal prediction model. This is however not our intention. We rephrased respective sentences.

- R1 – comment 12:
  Discussion and Conclusions: this section is too long, consider shortening. Parts of the discussion may be moved to the Results. The last sentence of the article is not clear, please revise.
  **Response to R1 – comment 12**:
  Last sentence was changed to: "An important question, that should be addressed in future studies is: Which physical processes form the foundation of this detected decadal prediction skill for the different circulation variables?"

- R1 – comment 13:
  Fig4: significant at what level
  **Response to R1 – comment 13**:
  We added "(95% significance level)" to p.5 l.10.

- R1 – comment 14:
  p2,l.11: remove comma before dash. p2, l18-19: put references in brackets p2,

l29: did you mean more skilfull ? Skilful is misspelled. p5, l11: 1000 time - should '1000 time steps' be better? p10, l6: should read 'these results'

**Response to R1 – comment 14**:

We replaced the dash with a comma, to separate the two independent but related sentences.

Thank you for noticing, brackets were inserted.

No, we do mean simply skillful.

We are using American English throughout the paper, the spelling is correct: skillful.

We use the bootstrap method, to estimate the distribution of a population by resampling the dataset with replacement. This is repeated 1000 times. The suggested term "1000 time steps" is not applicable.

The word "results" is used as a verb here. The suggested use of the word "results" as a noun would change the meaning of the sentence and leave it incomplete.
* * *
**Strength of North Atlantic Cyclones**

Figure: Bar chart titled "Strength of North Atlantic Cyclones" with x-axis "Laplace p" ranging from 0.7 to 4.45, and y-axis "cyclones per winter" ranging from 0 to 60. Three systems are shown in the legend: ERA, HR, LR.

**Fig. 1.**

**Lifetime of North Atlantic Cyclones**

Fig. 2.

**percentage of grid points with signif. correlation to ERA−Interim**

Fig. 3.

---

## Author Comment (AC2) · 10 Jul 2019

Response to Comments of Anonymous Referee 2 [R2]

- R2 - comment 1:
  The applied methods are often not clear. The use of an "evaluation software" is mentioned (P5L3). What does it actually do? When is the ensemble mean calculated, e.g. are the shown correlation maps means of correlations or correlations between ensemble mean and reference. Please provide clarification and add the

applied calculation methods/equations. Could be as appendix/supplement.

**Response to R2 - comment 1**:

The evaluation software, as described in p.7, l.6-11, comprises the different post-processing routines to derive the stormtrack and the three different frequencies from the direct model output and it also comprises a routine for the skill (anomaly correlation) analysis. This evaluation software named "freva" was designed within the MiKlip project and used as Central Evaluation System by research groups within this project. Based on standardized model output, the "freva"-user can apply different evaluation or post-processing methodologies in an easy and reproducible way. What these single post-processing routines - or plugins as they are also called - do, is described in Section 2.2. This means, from the direct model output of the hindcasts, first the four circulation metrics and winterly averages of their statistics are calculated for the reanalysis and the hindcasts. Afterwards, lead time dependent anomalies and the anomaly correlation are calculated as follows: For each of the initialization experiments (1978, 1979, ...) the ensemble average (5 members) of the temporal mean of the 4 contained lead winters is calculated per grid point. This forms a new ensemble mean time series of the lead winters 2-5. This time series serves to calculate the climatology (temporal mean) and to calculate the respective anomaly time series. The time series of those anomalies of the hindcasts is then correlated (Pearson) to the time series of anomalies of the reanalysis. In decadal prediction studies, this procedure is usually repeated for each lead time, thus lead year 1, lead year 2-5, lead year 6-9 - it is therefore referred to as lead time dependent anomaly correlation. In our study we only show results for one lead time: lead winters 2-5.

Hence, the correlation maps in Fig. 4 and Fig. 5 show correlations between the ensemble mean and the reference.

We implemented this description to the manuscript text - see R1 comment 3.

- R2 - comment 2:

The study suggests a direct relation between the mean bias of the ensemble mean and the anomaly correlation of the ensemble mean to the reference for one and the same diagnostic. The correlation is insensitive to the mean bias on grid cell level, hence anomaly. It appears large parts of the result section and conclusions are based on the assumption that a reduction of mean error/bias leads to higher anomaly correlations for the same analyzed quantity. This has to be revised substantially.

**Response to R2 - comment 2**:

Thank you for the remark. It was not our intention to suggest a direct relation between bias and the anomaly correlation. Rather, the independent, but locally coincident, improvement of both, the bias and of the anomaly correlation, for the same quantity points towards an improvement of the physical processes in the HR model. We assumed we had already chosen our wording carefully. We revised and clarified the respective paragraphs.

• R2 - comment 3:

The hindcasts are presumably not post processed, e.g. corrected for time-varying bias, trend-adjusted, etc? Please clarify and state why this might be not necessary. Why is the approach of correcting biases of this study different to Kruschke et al?

**Response to R2 - comment3**:

In the third paragraph of Sec 2.1, we give information about how data is post-processed and analyzed. This is apparently not clear enough. Thank you for the comment.

We analyzed the frequencies of the circulation metrics, i.e. values for each lead winter, respectively, following the DCPP recommendation. That means that we calculated lead time dependent anomalies of those frequencies (see R2 comment 1 and R1 comment 3). This is a simple and robust approach to account for a possible lead time dependent mean bias, i.e. drift (DCPP recommendation,

Boer et al., 2016).

There exist miscellaneous more sophisticated approaches for the post-processing of decadal predictions (Kharin et al., 2012, Kruschke et al., 2016, Pasternack et al., 2018). In our study we wanted to point out the effect of the model resolution on the forecast skill of the circulation measures and therefore, we intentionally did not compare the LR model including a complex post-processing approach with the HR model including a complex post-processing approach.

- R2 - comment 4:
Spatial resolution has been discussed to be a serious limiting factor to correctly reproduce climate mean state and variability in the context of decadal prediction (e.g. Hewitt et al. BAMS 2017, Smith et al. QJRMS 2016). This should be mentioned more prominently in the motivation and put in context of this study in the discussion. There are numerous studies about the effect of resolution in climate models in general including the effect on North Atlantic circulation measures (e.g. Davini et al. J ADV MODEL EARTH SY, 2017). How do they compare to this study?

**Response to R2 - comment4**:
We dedicated an entire paragraph of the introduction (p.3, l.1ff) to this topic. We discussed the limiting factor 'resolution of climate models' and its effects on the representation of the ocean surface state and in particular on the representation of the North Atlantic atmospheric circulation, and we cited a multitude of studies dealing with this topic. Also, we have discussed state of the art results from studies in which higher resolved decadal hindcast sets were analyzed. Nevertheless, we added some of the suggested papers to our citation list, where appropriate.
Added before p.3, l.1:
"It is well known that a coarse spatial resolution of global coupled climate models hinders the proper representation of sub-synoptic scale systems, and thus the

climate mean state and variability."
Added to p.3 l.10:
"Similar effects for the blocking frequency bias are found in an atmosphere only model by Davini et al. (2017)."
Added Hewitt (2016) to paper listing:
"It has been found in many studies, that the atmospheric dynamics benefit not only from a coupling of the atmosphere and ocean but also from an increased model resolution (Shaffrey, 2009; Jung 2012; Dawson, 2013; Hewitt 2016)."
Added to p.14, l.23:
"A similar change in blocking frequencies with increased model resolution was also found in Davini et al. (2017)."

- R2 - comment 5:
  The difference in mean bias for LR and HR in cyclone frequency is striking and given the very small differences in stormtrack activity somewhat unexpected, e.g. at 30W, 50N (Fig 2a vs Fig 3a). The result is apparently similar to Kruschke et al 2014. Kruschke et al compare uninitialized experiments in LR to NOAA's 20th Century Reanalysis. In their study the mean bias is up to 25 systems per winter over the North Atlantic and they mention a possible underestimation of cyclone frequency of the reanalysis. This seems at odds with what is shown here: A mean bias of up to 80 systems and more per winter in comparison to a different reanalysis product. Please discuss this. Is it possible to estimate how much is due to the applied tracking method? One suggestion could be to interpolate the HR hindcast to the lower resolution and repeat the analysis. Will that change the results? This could be done for a single member and put as appendix. It is mentioned that LR overestimates weak and moderate systems. Why?
  **Response to R2 - comment 5**:
  Regarding your suggestion to interpolate HR to the lower resolution: Usually the experience is, that the finer the resolution of the model, the more accurate the

description of the pressure field and the more cyclones can be detected by the algorithm. So, if we interpolated HR to the lower resolution, we would not expect to see an LR-like positive bias - rather the opposite is the case, we would expect even less cyclones in the interpolated HR hindcasts. This would not be helpful to explain the strong positive cyclone bias. We therefore decided not to follow this suggestion. We understand, that your question points towards an explanation for the strong positive cyclone frequency bias in LR. This question is already partly answered in the response to R1 comment 7 (positive cyclone frequency bias is produced by weak and short-lived cyclones) and is complemented by the next few paragraphs.

The cyclone identification and tracking method applied in our study is identical to the one used in Kruschke et al. (2014) - the methodology originally designed by Murray and Simmonds (1991) - so the differences in cyclone frequency biases in the two studies cannot be derived from a different methodology. The same holds for the computation of the frequencies, in both cases the frequency was derived from cyclone counts within a distance of 1000 km around a grid point. The differences can however be explained by the different datasets used. In Kruschke et al. (2014) the bias of the un-initialized LR runs (of an older MPI model version) relative to 20CR is shown. In our study the bias of the initialized LR runs (of the current MPI model version) relative to ERA-Interim is shown - so the MPI model version differs, the initialization differs and the reanalysis dataset differs. We performed a few studies in the attempt to isolate the different effects.

Effect of the new model version

To test the influence of the model development on the bias, we analyzed the cyclone frequencies in the un-initialized MPI-ESM runs used and shown in Kruschke et al. (2014) and in the respective un-initialized runs of the MPI-ESM model used in our study - please note, that we never showed results from the un-initialized runs, but only from the initialized runs in our paper.

This model development from the system used in Kruschke et al. (2014) termed

'Baseline1' (B1) to the current MPI-ESM system termed 'Pre-operational' (Preop) slightly reduces the winterly cyclone counts over the North Atlantic (review response Fig. 4). The effect is negligible (-4 cyclones per winter) compared to the strong bias we see in the initialized Preop-LR, and is of opposite sign. Thus, the model development alone cannot explain the strong North Atlantic cyclone frequency bias.

Effect of the initialization

The comparison between the initialized Preop-LR runs used and shown in our study and the respective un-initialized runs of the Preop-LR system however shows a very strong increase in North Atlantic cyclone frequencies (+100 cyclones per winter; review response Fig. 5). This indicates that the majority of the bias seen in Fig. 3a of our study can be explained by the initialization of the Preop-LR system.

Actually, this initialization effect is also inherent in the older B1 system (review response Fig. 6), between the un-initialized runs used and shown in Kruschke et al. (2014) and the respective initialized runs of the same system also used in Kruschke et al. (2014) - but they only showed the bias for the un-initialized runs in their paper.

Given the fact that the initialization technique in Preop-LR and Preop-HR is identical, but only LR exhibits the strong cyclone frequency bias, it appears to be an unfavorable interaction, between the LR system and the initialization, which triggers this bias. In the following we explored what this interaction might entail.

Taking a closer look into the initialized LR system, we find a negative sea-level-pressure bias over the central North Atlantic. This is shown in the review response Fig. 7 (left) for the initialized simulations used in our study; and for the un-initialized simulations of the same model version in Müller et al. (2018, their Fig. 7c). The systematically too low pressure over the central North Atlantic seems to affect existing flow disturbances, i.e. weak/open cyclones, over the central North Atlantic, by strengthening them and artificially extending their

lifetime just enough to meet the algorithm's thresholds, so that a strong bias in the average cyclone frequency becomes visible. As shown in the intensity and lifetime histograms, in response to R1 comment 7, this bias can be attributed to weak and short-lived cyclones. Obviously this pressure bias in LR acts to produce artificial cyclones.

Although a negative pressure bias is still visible in HR (review response Fig. 7, right) but shifted to Newfoundland, we do not see a likewise strong bias in the cyclone frequencies there. The negative pressure bias in the cyclogenesis area (Newfoundland) seems not to be as critical. We conclude, that the negative pressure bias in the two hindcast systems is more relevant for existing disturbances (strengthening those to become weak and moderate cyclones over the North Atlantic in LR) than for the genesis of cyclones (over Newfoundland in HR).

Effect of the reanalysis

To round off the picture, we compared the cyclone track density biases of the initialized and un-initialized MPI systems relative to different reanalysis datasets. The plots in review response Fig. 8 are for the B1 system, but they look essentially identical for the Preop-LR system. The bottom, left figure corresponds to the bias seen in Kruschke et al (2014) - a bias of 20-30 cyclones over the Eastern North Atlantic and Europe for the un-initialized system relative to 20CR. If they had used ERA-Interim instead of 20CR the top, left figure would have appeared - a general underestimation of the un-initialized B1 system over the Northern North Atlantic of -20 to - 40 cyclones. The comparison between the left and right column illustrates again the initialization effect. The top, right figure is equivalent to the bias shown in our study - a bias of +80 cyclones over the central North Atlantic in the initialized system relative to ERA-Interim.

- R2 - comment 6:
  The ensemble spread is unfortunately not used or shown for any of the analy-
ses. How is the spread different between LR and HR? Is the reanalysis within the spread?

**Response to R2 - comment 6**:

Instead of checking whether the reanalysis is within the spread, we follow the CMIP or DCPP suggestion to compare the ensemble spread with mean squared error of the model compared to the reanalysis - to see if the spread is an adequate representation of the uncertainty. The spread is equally strong in LR and HR and close to the MSE (applying the Log. Ensemble Spread Score) for each of the respective quantities (stormtrack, blocking frequencies and windstorm frequencies - not shown). For those quantities it is not necessary to show the plots. Only for the cyclone frequencies (review response Fig. 9), the spread in LR is larger than in HR over the North Atlantic, i.e. where the bias is high, and over Eastern Europe. This means that additionally to the average cyclone bias, created by the North Atlantic pressure bias and the initialization (as discussed in response to R2 comment 5), the members produce largely varying numbers of cyclones per winter. This result is in agreement with the bias in weak cyclones as shown in R1 in comment 7. Still, the ensemble spread in LR is not overwhelmingly high. We added two sentences to the manuscript.

- R2 - comment 7:

When analyzing absolute numbers (here for blocking, cyclones and windstorms) ties have to be considered in the correlation calculation, ie seasons with the same number of events. Presumably ties are not taken into account as the manuscript does not mention it. 2 possible solutions: i) mask regions with a large number of ties ii) use a different correlation coefficient, e.g. Kendall's Tau B. Otherwise the correlation value could be misleading and statistical significance becomes meaningless, especially in regions with few events per season. There is a significant negative correlation in windstorm frequency in LR over Eastern Canada and a significant positive correlation in HR over the same region. This could be an example of too many ties.

**Response to R2 - comment 7**:

In the manuscript there is a lag in the explanation of how the time series are pre-processed before correlations are calculated. Thank you for this feedback! We added a more detailed description to Sec. 2.1 where this is explained. It includes the information of anomaly, ensemble mean and running mean computations. Due to this type of preprocessing we decided to use the Pearson correlation co-efficient instead of rank correlations - the latter would have been affected by ties. Nevertheless, we analyzed the number of ties and found, that due to the ensem-ble mean and running mean, there are almost no ties in the hindcasts, and only few ties in the reanalysis data.

Significance of the correlation is calculated by means of a bootstrapping, resam-pling the time series with replacement. Ties (only few cases as mentioned) are also used for the bootstrapping which leaves significance still meaningful.

R2 - comment 8:

Related to the above point: Cyclone frequency is apparently masked in regions with high orography. This can be seen in Fig 5. Why is there no mask in Fig 3? What about wind storms. Why are windstorms not masked? Please also con-sider masking regions with few events per season. There is a mask for blocking. Please state why.

**Response to R2 - comment 8**:

Thank you for the remark, there should indeed be a mask for cyclones in Fig. 3, we updated the figure. The reason why cyclone frequencies are masked, is because they are derived from the mean-sea-level pressure. Over higher terrain, this quantity has to be extrapolated from the elevated surface pressure to sea-level. This extrapolation is inaccurate over very high terrain which would lead to the identification of artificial cyclones, therefore cyclones identified over those areas are excluded from the tracking (Murray and Simmonds, 1991). The wind-storms, however, are computed from the 98th percentile of surface wind speeds,

which is not influenced by high terrain. Therefore, the windstorm frequencies need no mask. For the blocking, there was no mask used. As explained in chapter 2.2 (p.6, l.7) anticyclones are only identified between 35° and 80°N. For this quantity, subtropical regions are usually excluded from blocking identification analyses to avoid the influence of the subtropical belt of high pressure systems.

- R2 - comment 9:
  The discussion is not critical enough. The reader gets the overall impression of a nearly perfect prediction system regarding the analyzed quantities. Mentioning the correlation value could sometimes already be enough to put the results in perspective. There are some inconsistencies as mentioned above that should be discussed. There is only one sentence P16, L15ff with reference to previous studies with similar objectives. Please add some references or state the lack thereof. See also point 4)
  **Response to R2 - comment 9**:
  We acknowledge that we have strongly emphasized the positive effects of the increased model resolution, partly at the expense of fair balance. We thoroughly double checked the discussion and rephrased expressions that could lead to the impression that HR is the perfect prediction system.
  We stated in the introduction (p.3 l.21), that our study is the first that explores the effects of model resolution on the decadal prediction skill of extratropical circulation metrics. However, we now added this information also to the discussion and inserted the following to p.16 l.17:
  "To this date there is no study that addresses the effect which the model resolution has on the decadal prediction skill on extratropical circulation metrics. However, our results are in agreement..."

- Minor comments: i) The title suggests an analysis of the entire NH. Please correct. Consider adding the word "deterministic" in the title
  **Response**: Thank you for the remark - we changed the title to "Improvement in

the decadal prediction skill of the North Atlantic extra-tropical winter circulation through increased model resolution"

- ii) "Anomaly correlation" and "skill" are used as synonyms throughout the manuscript. Please state that deterministic skill is assessed through anomaly correlation somewhere in the paper and in the abstract.
  **Response**: "Significant positive anomaly correlation" and "Skill" are used as synonyms. This is stated at p.10 l.30, and a respective note was added to p.1 l.6: "The deterministic predictions are considered skillful, if the anomaly correlation is positive and significant."

- iii) There is no reference for the "common shortcoming of climate models" of a too zonal stormtrack in the introduction.
  **Response**: The reference Scaife et al. (2011) was added to p.3 l.5.

- iv) P1L1: The acronym MiKlip is not explained
  **Response**: The full name for the acronym was added to p.2 l.10.

- v) P1L8: "functional chain" is not clear
  **Response**: Replaced "functional chain" with "chain".

- vi) P1L11ff: Newfoundland is not "downstream" of the stormtrack.
  **Response**: Newfoundland is enumerated together with Central Europe, those are the regions where the windstorm frequency improves. Central Europe is downstream of the stormtrack. Newfoundland is mentioned for reasons of completeness. Though the formulation is imprecise it is not wrong. We added "primarily" to the preceding sentence, to improve precision.

- vii) P1L20ff: Please add reference for this paragraph
  **Response**: We added: Leckebusch2004, Ulbrich2009, Sillmann2009, Pfahl2012, DeutscheRueck2018
- viii) P2L8: "sectors" is most likely the wrong word
  **Response**: Replaced "sector" with " division".

- ix) P2L20ff: restructure sentence: "One result..."
  **Response**: Sentence was restructured.

- x) P3L1: specify "lower resolution"
  **Response**: about 1.5° horizontal grid spacing or less

- xi) P3L8: "functional chain?"
  **Response**: Replaced "functional chain" with "chain".

- xii) P3L22: change "variables" to "diagnostics" or similar. Variable is not the correct word.
  **Response**: Thank you for this note. We replaced "variable" with either "quantity" or "diagnostic" in various positions of the manuscript.

- xiii) P3L29: same, please check the entire manuscript
  **Response**: see above

- xiv) P4L12ff: "However...": Please rephrase
  **Response**: Rephrased to: "However, there exists no gridded observational dataset for the metrics that we analyze."

- xv) P5L1: add "deterministic". See points i) and ii)
  **Response**: We added "deterministic".

- xvi) P3L2: centered or uncentered anomaly correlation? See point 1)
  **Response**: The definition of the centered and uncentered anomaly correlation, e.g. as in Wilks' "Statistical Methods in the Atmospheric Sciences", refers to spatial correlations, i.e. of pairs of grid points in the observed and forecast fields.

However, in our study, as described in response to R2 comment 1 and R1 comment 3, we apply a temporal correlation (Pearson) of anomalies for each individual grid point. In order to avoid misunderstandings, we could have changed the expression from "anomaly correlation" to some other, more distinct and probably longer term. But we decided to keep it like that, to be conform with previous studies of the MiKlip decadal prediction system, that also used the term "anomaly correlation". Also, we think the updated and very detailed description of our evaluation procedure is clear enough to avoid a misunderstanding.

- xvii) P6L32ff: This is unclear and probably wrong somehow. What kind of percentile is used? Is it the same one in LR and HR? This might explain why the difference in cyclone frequency is not apparent in windstorms
  **Response**: To be more clear we refined wording and replaced hindcast by model simulation. The explanation is correct. For each simulation (LR, HR, reanalysis), a different threshold is used, i.e. the local percentile of the individual simulation. This is a feature of the algorithm which implicitly adjusts means bias. The idea of the methodology is explained by Leckebusch et. al (2008). To calculate model consistent percentiles, uninitialized simulations of LR and HR are used as done by Kruschke et al (2016).

- xviii) P7L2ff: change "nicely illustrated"
  **Response**: Changed to "demonstrated".

- xix) P7L31: the value in brackets is easily misunderstood. Maybe: -3% of a total of X% days in one season
  **Response**: Added unit information to p.7 l.31: "The blocking frequency shows a strong negative bias of fraction of blocked days per winter (-3%) in the LR system"

- xx) P10L18: change "implying" to "could be due to" or similar
  **Response**: Changed to "possibly due to".

- xxi) P12L4ff: see point 2 for the whole paragraph
  **Response**: Revised where needed.

- xxii) P12L22ff: see point 2
  **Response**: We assume you mean page 13 instead of page 12. In P13L22 we simply state that areas of skill improvement coincide with areas of bias improvement. There is no description of dependency or of cause and effect.

- xxiii) P13L35ff: improvement in cyclone frequency improves windstorm frequency? Specifically along the European western coast? P10L12ff highlights the differences of the 2 diagnostics
  **Response**: The differences between windstorms and cyclones explained in P10L12ff refer to the positive cyclone frequency bias over the central North Atlantic, which is caused (as we had suggested in the submitted draft and now proved in the review process) by weak and short-lived cyclones. The argumentation in this paragraph is used to clarify that a rather weak bias in the windstorm frequency is not at all contradictory to the strong cyclone frequency bias, because the windstorms can be considered a subset of the cyclones, and the other subset which is not equivalent to the windstorms (i.e. the weak cyclones) can explain the positive cyclone frequency bias. This explanation is not contradictory to the fact that the skill in cyclone frequency affects the skill of the windstorm frequency (P13L35ff), because still the cyclone frequency covers all intensities of systems, those that do and those that do not produce storms. It is therefore possible and likely, that the subset of strong cyclones influences the windstorm frequency and its skill, respectively.

- xxiv) P15L8: Muller et al 2018 show a decrease of MSLP bias in the Eastern North Atlantic but an increase in the Western North Atlantic in HR. It is therefore only partially "in line".
  **Response**: We added "over the central North Atlantic" to be more precise.

- xxv) P15L25ff: see point 2, for blocking + cyclones
  **Response**: Again, we only say the areas of skill and bias improvement coincide. We rephrased the second part of the sentence.

- xxvi) P5L10: Please provide a reference or calculation method for the statistical significance. Is the calculation method different between correlation significance and significance for the differences in correlation
  **Response**: A reference was added (Goddard et al., 2013) to p.5 l.4.

[Figure]

[Figure]

**Fig. 4.** Effect of the model development - Difference of the cyclone frequency between the un-initialized Preop-LR and un-initialized B1-LR simulations

[Figure]

**Fig. 5.** Effect of the initialization in Preop-LR - Difference of the cyclone frequency between the initialized Preop-LR and un-initialized Preop-LR simulations

[Figure]

[Figure]

**Fig. 6.** Effect of the initialization in B1-LR - Difference of the cyclone frequency between the initialized B1-LR and un-initialized B1-LR simulations

[Figure]

**Fig. 7.** Mean Sea-Level Pressure bias relative to ERA-Interim - left: in the Preop-LR system;
right: in the Preop-HR system

[Figure]

**Fig. 8.** Cyclone frequency bias in the different simulations relative to different reanalyses - top: relative to ERA-Interim; bottom: relative to 20CR; left: un-initialized simul.; right: initialized simul.

[Figure]

**Fig. 9.** Spread vs. MSE (Logarithmic Ensemble Spread Score - LESS) for the cyclone fre-
quency - left: in the Preop-LR system; right: in the Preop-HR system

---

## Referee Report (RR1)

General comment:
Thank you for your thorough answers and explanations. The applied methods are clearer now. I have only one comment regarding my previous comment 5 about why there are more cyclones in LR.

Recommendation:
Acceptance after minor revisions

previous comment 5)
Thanks for your answer to this comment and the additional figures. This large bias reduction from LR to HR only in initialized predictions compared to a neglectable bias reduction from LR to HR in free running experiments is surely an interesting result. It is important for the reader to know that the reason why there are so many more cyclones in LR is due to initialization. That's why I strongly encourage to add your figure 5 from the responses (Initialized Pre-Op minus un-initialized Pre-Op) to the manuscript.

The question why this large bias in LR remains.
The provided explanation of "systematically too low pressure over the North Atlantic" is not sufficient. Lower pressure at the downstream end of the stormtrack goes along with more cyclones.

Please either
A) comment on why the difference in LR between initialization and no-initalization will not be further investigated in this study

B) discuss potential reasons. This will require additional analyses.
The most obvious reason seems to be related to the role of the ocean in the gulf stream region. Note also the differences in stormtrack and windstorm frequency between LR and HR in this region.
The focus of the manuscript is forecast years 2-5. Looking at the differences in drift between LR and HR both in SST and cyclone frequency might help.

Technical comments:

i) P7L33: "blocking_2d/detail" should be "blocking/detail"
ii)P11L3: delete "and Canada"
iii) P11L11ff: when including the above mentioned figure this paragraph has to be revised
iv) P11L26: replace "eligible"
v) P16L15: correct "persits"

---

## Author Response (AR2)

**Author's Response to Minor Revisions**

M. Schuster

October 15th 2019

**Contents**

| 1        | Point-by-point response to the reviews                                        | 2 |
|----------|-------------------------------------------------------------------------------|----------|
|          | 1.1 Response to Referee Report $\#1$ [R1]                                     | 2        |
|          | 1.2 Response to Referee Report $\#2$ [R2] $\ldots$ $\ldots$ $\ldots$ $\ldots$ | 4        |
| 2 | Marked-up manuscript version                                                  | 6        |

Thank you to both reviewers for your remarks. We have incorporated your suggestions in the manuscript. Please find the point-by-point response below. The most important amendment we have made is the addition of a supplement. In this supplement we included the figure suggested by the second reviewer, as well as other figures explaining the initialization effect that we detected in our study. Also, one citation changed, which does not show up in the marked-up manuscript, the respective paper is meanwhile in review in a different journal (now: Kadow et al., Freva - Free Evaluation System Framework for Earth System Modeling, Journal of Open Research Software, in review).

**1 Point-by-point response to the reviews**

**1.1 Response to Referee Report #1 [R1]**

Regarding the comment on the length of the discussion, we split this section into two subsections - one on the bias and one on the prediction skill, analogue to section 3 "Results".

Minor comments of referee R1 were:

- p.1,l. 3: remove "as well as" **Response**: Removed.
- p1,l.5: remove "each"; say "in ensembles of 5 members in ..." **Response**:

Changed to "... evaluated in ensembles of 5 members in a lower... and a higher resolution version"

• p2, 11-3 : "fluctuations in the circulation and thus i.a. the frequency of extremes such as strong cyclones, intense windstorms or phases of blocked flow " this is hard to read, consider revising

**Response:**

Inserted "in" and a comma: "...fluctuations in the circulation and thus i.a. in the frequency of extremes, such as strong cyclones, intense windstorms or phases of blocked flow."

- p2,l5 : remove "and harm" **Response**: Removed.
- p2, l9: remove "itself" **Response**: Removed.
- p2,127: "which exists irrespective of the ocean initialization technique (Moemken et al., 2016) " either remove or rephrase the sentence **Response**:

We do not see, why this sentence would need rephrasing. It clearly explains, that the skill for the different quantities is found independent of the initialization technique. This information should not be withheld. Therefore, we neither rephrased nor removed the sentence.

• p2, l33: "the European North Sea " - I am not aware of a non-European North Sea

```
Response:
Removed "European".
```

- p.7, l8: Laplace?
  Response: Changed to "Laplacian of pressure".
- p.11,l24: capable of representing **Response**: Adopted.
- p.11,l.26: I'd recommend "dynamics ... benefits" **Response**:

The current spelling is correct, we consulted a native speaker on this matter: "The dynamics benefit from the increase in resolution." We did not add the suggested "s".

• p.16, l15: "persits" should read "persist"; "the North Pacific blocking, the North Atlantic blocking"

Response:

Changed to: "... the underestimation ... mainly persists ..." and "... in contrast to North Pacific blocking, the North Atlantic blocking ..."

• p.16, l27L: "The proportion of the bias of a) the different reanalysis used in Kruschke et al. 2014 and b) the model physics, that has since been further developed, is negligible (not shown). " - review this sentence **Response**:

Changed to: "The contributions to the bias through a) the different reanalysis used in Kruschke et al. (2014) (Fig. S3) and b) the different model physics of the meanwhile advanced prediction system (Fig. S4) are negligible."

p16,l33: "This bias cannot be corrected ..." - sounds as if there was some obvious reason for that. Either explain or rephrase.
 Response:

Changed to: "This bias is not corrected..."

• p.17, l9: "a strong and significant skill " - strong is most likely significant. If you still want to emphasise significance I suggest: strong (statistically significant) skill.

Response:

Adopted.

- p.17,l17: non-existent **Response**: Adopted.
- p.19,l7: line of arguments

**Response:**

The current spelling without "s" is correct - we consulted a native speaker on that matter.

• p.19,l16: remove "but linked"

**Response:**

With this term we would like to emphasize that the different quantities that we analyzed are physically linked and therefore contribute to understanding the processes of the skill improvement. We did not remove the term but instead added "physically" for clarification.

• Conclusions: I suggest mentioning a better representation of cyclone characteristics in HR version (shown in fig.4), not just their frequencies **Response**:

We added a sentence to the abstract: "Especially cyclones, i.e. their frequencies and characteristics like strength and lifetime, are better represented in HR."

**1.2 Response to Referee Report #2 [R2]**

• previous comment 5)

Please either

A) comment on why the difference in LR between initialization and noinitialization will not be further investigated in this study

B) discuss potential reasons. This will require additional analyses.

**Response:**

The effect of additional and artificial short-lived and weak cyclones that only occurs when the LR system is initialized was detected during this study. It is a rather technical issue and model developers will surely be interested in this feature. The first couple of analyses which we had performed to isolate the source of this behaviour, e.g. the analysis of biases in basic variables such as sea surface temperatures and mean sea level pressure, had not revealed a clear picture yet of a process that would only be found in LR but not in HR. Thus, a more detailed analyses and maybe even the setup of a new hindcast set (e.g. ocean initialization only vs. atmosphere initialization only) would be necessary which are however beyond the scope of this paper. Therefore we added a respective sentence to Sec. 4.1 - the discussion on the model bias. (see point iii below)

Technical comments of referee R2 were:

• i) P7L33: "blocking\_2d/detail" should be "blocking/detail" **Response**:

No, in fact there are two blocking plugins available in the evaluation system freva, one for one-dimensional blocking (instantaneous blocked longitude) and one for two-dimensional blocking. We used the latter. Our current description is correct.

- ii)P11L3: delete "and Canada" **Response**: Adopted.
- iii) P11L11ff: when including the above mentioned figure (figure 5 from the responses: Initialized Pre-Op minus un-initialized Pre-Op) this paragraph has to be revised
  - **Response**:

We agree that it is of interest for the reader to know more about the origins of the positive cyclone frequency bias. However, if we included Fig. 5 from the responses to the results of the main manuscript, we fear that this would shift the focus to a rather technical aspect which also is not fully understood yet. It would need further investigations beyond the scope of this work to explain what exactly triggers the bias. Still, it is of importance that this initialization effect was detected during our study and it deserves to be mentioned. We therefore decided for a compromise: Figures from the responses are now included in a separate PDF document (supplement) - to illustrate the different effects of the initialization, the different reanalyses and the model development. These effects were already discussed in Section 4 in the previous version of the manuscript (p.16 1.22) - instead of "(not shown)" the respective passages now refer to the corresponding figures in the supplement.

• iv) P11L26: replace "eligible" **Response**:

Changed to: "suitable"

• v) P16L15: correct "persits" **Response**: Changed to: "persists" 2 Marked-up manuscript version

**Improvement in the decadal prediction skill of the North Atlantic extra-tropical winter circulation through increased model resolution**

Mareike Schuster1, Jens Grieger1, Andy Richling1, Thomas Schartner2, Sebastian Illing1, Christopher Kadow1, Wolfgang A. Müller4, Holger Pohlmann3,4, Stephan Pfahl1, and Uwe Ulbrich1

1Freie Universität Berlin, Institut für Meteorologie, Carl-Heinrich-Becker Weg 6-10, 12165 Berlin

2Deutscher Wetterdienst, Güterfelder Damm 87-91, 14532 Stahnsdorf

3Deutscher Wetterdienst, Bernhard-Nocht-Straße 76, 20359 Hamburg

[revised manuscript text omitted]